# KABB: Knowledge-Aware Bayesian Bandits
# for Dynamic Expert Coordination in Multi-Agent Systems

**Jusheng Zhang** [* 1]   **Zimeng Huang** [* 1]   **Yijia Fan** [1]   **Ningyuan Liu** [1]

**Mingyan Li** [1]   **Zhuojie Yang** [1]   **Jiawei Yao** [2]   **Jian Wang** [3]   **Keze Wang** [1 4 5]

## Abstract

As scaling large language models faces prohibitive costs, multi-agent systems emerge as a promising alternative, though challenged by static knowledge assumptions and coordination inefficiencies. We introduce Knowledge-Aware Bayesian Bandits (KABB), a novel framework that enhances multi-agent system coordination through semantic understanding and dynamic adaptation. The framework features three key innovations: a customized knowledge distance model for deep semantic understanding, a dual-adaptation mechanism for continuous expert optimization, and a knowledge-aware Thompson Sampling strategy for efficient expert selection. Extensive evaluation demonstrates KABB achieves an optimal cost-performance balance, maintaining high performance while keeping computational demands relatively low in multi-agent coordination.

## 1. Introduction

With the rapid advancement of large language models (LLMs), their applications have expanded to complex tasks such as cross-domain knowledge integration and multistep decision-making. Although many LLMs (Achiam et al., 2023; Liu et al., 2024; Adams et al., 2024; Team et al., 2024; Bai et al., 2023) demonstrate impressive versatility in various tasks through techniques such as in-context learning and instruction-tuning, their performance remains constrained by factors such as model size and the limitations of training data (Jiang et al., 2023; Lu et al., 2024a; Zhang et al., 2025). Scaling these models further to improve performance

is prohibitively expensive and often requires retraining on datasets comprising trillions of tokens.

Multi-Agent Systems (MAS) (Guo et al., 2024) offer a promising alternative by coordinating multiple specialized agents to achieve superior performance compared to individual systems while maintaining manageable computational costs and budgets. Recent advances in MAS have led to the development of several frameworks. For example, the Mixture of Agents (MoA) (Wang et al., 2025) employs multiple LLMs as proposers to iteratively refine responses, with a central aggregator delivering the final output. Although MoA has demonstrated robustness and scalability in deployment, its computational cost scales linearly with the number of agents, and significant redundancy and noise become a problem. For example, on datasets like MATH (Hendrycks et al., 2021), weaker models in the ensemble often interfere with the aggregator's decisions, leading to incorrect results (see Figure 1).

Alternatively, Mixture of Experts (MoE) frameworks (Gong et al., 2024; Zhang et al., 2024; Wang et al., 2024a; Tang et al., 2024), in the context of multi-agent systems, focus on fostering collaboration among domain-specific experts, enabling the integration of diverse responses across fields. This approach reduces redundancy and noise, but is often limited to predefined tasks. A fundamental limitation of both frameworks lies in their reliance on static knowledge assumptions, making them ill-suited to address dynamic changes in expert capabilities or the emergence of novel concepts. These limitations highlight deeper challenges in MAS, particularly in areas such as knowledge understanding, response integration, and dynamic adaptability.

The increasing complexity of real-world scenarios requires systems that can adaptively select relevant knowledge domains and identify the optimal combination of experts. Multi-Armed Bandit (MAB) algorithms (Mahajan & Teneketzis, 2008) have emerged as a powerful tool for tackling such dynamic decision problems. By striking a balance between "exploration" (discovering new expert combinations) and "exploitation" (leveraging known successful strategies), MAB can continuously optimize system perfor-

---

[*]Equal contribution  [1]Sun Yat-sen University [2]University of Washington [3]Snap Inc. [4]Peng Cheng Laboratory [5]Guangdong Key Laboratory of Big Data Analysis and Processing, Guangzhou. Correspondence to: Keze Wang <kezewang@gmail.com>.

*Proceedings of the 42nd International Conference on Machine Learning*, Vancouver, Canada. PMLR 267, 2025. Copyright 2025 by the author(s).

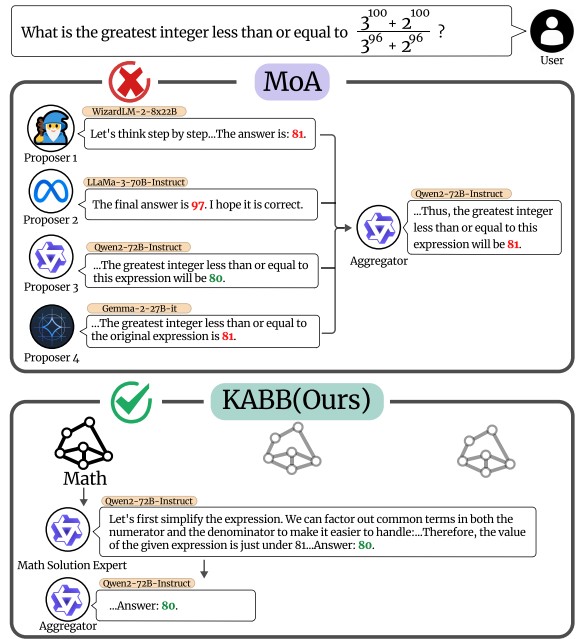

*Figure 1.* Comparison of MoA and KABB (Ours) on solving a mathematical problem: MoA's aggregator is misled by conflicting weaker proposals, resulting in an incorrect answer, while KABB employs a knowledge-aware approach to drive related experts and arrive at the correct solution.

mance. However, traditional MAB approaches rely solely on historical feedback(Diao et al., 2025), often overlooking the semantic relationships between tasks and experts.

To bridge this gap, knowledge graphs (Ge et al., 2024) provide a compelling framework for representing and leveraging these semantic connections. By structuring expert capabilities and task requirements as interconnected knowledge networks, knowledge graphs enable: (1) precise modeling of dependencies across knowledge domains, (2) dynamic tracking of expert capabilities over time, and (3) identification of knowledge gaps in task-solving pathways. This structured representation not only enhances the accuracy of expert selection but also provides semantic-level guidance for response integration. Together, these advancements pave the way for more adaptive, efficient, and semantically informed multi-agent collaboration systems.

In this work, we propose the Knowledge-Aware Bayesian Bandits (KABB) framework to significantly enhance the coordination capabilities of multi-agent systems through three core innovations. First, we introduce a customized knowledge distance model grounded in deep semantic understanding, which surpasses traditional keyword-based methods by integrating concept overlap, dependency path optimization, and dynamic historical performance evaluation. Specifically, expert capabilities and task requirements are represented as vectors, with concept overlap calculated using enhanced cosine similarity, dependency path lengths optimized through hierarchical knowledge relationships, and historical feedback dynamically adjusted via an adaptive time-decay factor. These components are unified into a comprehensive distance metric, further refined with deep learning techniques to optimize the weight parameters.

Second, we develop a dual adaptation mechanism to support continuous expert optimization and knowledge evolution. This mechanism employs Bayesian parameter updates with exponential time decay to mitigate the influence of outdated data while dynamically adjusting key metrics within the knowledge graph, such as concept overlap and historical performance. This ensures that expert capabilities remain adaptive to the evolving demands of tasks in real-time.

Finally, we design a knowledge-aware Thompson sampling strategy to improve computational efficiency in expert selection. By incorporating the knowledge distance metric into the Beta distribution sampling process, our strategy enables efficient identification of the top-k experts for dynamic decision-making. This approach demonstrated significant improvements in performance and cost efficiency on leading datasets like AlpacaEval 2.0 (Dubois et al., 2024). Additionally, a two-stage knowledge graph-guided response integration process ensures logical consistency by detecting semantic conflicts and enhancing contextual coherence, thus substantially reducing contradictory output.

Together, our innovations enable the KABB framework to effectively address the challenges of dynamic expert coordination, offering a scalable, adaptive, and semantically informed solution for multi-agent systems in complex real-world scenarios.

## 2. Related Work

### 2.1. Large Language Model Ensemble

The ensemble of large language models (LLMs) has emerged as an effective strategy to leverage the complementary strengths of different models and improve performance across diverse tasks. Early approaches primarily focused on combining outputs from multiple models through techniques like reranking or probability distribution averaging. For instance, Jiang et al. (2023) proposed PAIRRANKER for pairwise output comparisons and GENFUSER for generating improved responses by synthesizing multiple candidates. Similarly, Huang et al. (2024) explored output fusion by averaging probability distributions, while FrugalGPT (Chen et al., 2023) introduced a cost-efficient cascading mechanism that allocates tasks dynamically across LLMs to

reduce computational overhead. These methods highlight the potential of ensembling to amplify individual model capabilities while addressing computational constraints.

Beyond simple output aggregation, recent research has shifted toward more dynamic and adaptive frameworks for LLM collaboration. Mixture-of-Agents (MoA) (Wang et al., 2025) exemplifies this trend by introducing iterative refinement processes where multiple LLMs serve distinct roles, such as generating and refining responses through multi-layered agent interactions. This approach emphasizes the importance of both diversity and performance in model selection, demonstrating that combining heterogeneous models often yields superior results compared to homogeneous ensembles. Additionally, routing-based methods, such as those proposed by Wang et al. (2024a) and Shnitzer et al. (2023), optimize efficiency by dynamically selecting the most suitable model for a given input, while ZOOTER (Lu et al., 2024b) further refines this concept by distilling model expertise without requiring full inference for all candidates. These advancements highlight the progress in LLM ensemble techniques, focusing on efficiency and quality. Building on this, we propose a framework that integrates knowledge-aware mechanisms to improve adaptability and semantic coherence in multi-agent systems.

### 2.2. Multi-Armed Bandit for Decision Optimization

The Multi-Armed Bandit (MAB) framework balances exploration and exploitation in sequential decision-making under uncertainty. Classical algorithms like UCB and Thompson Sampling excel in recommendation and resource allocation, while Contextual Bandits and adaptive methods refine decision-making in dynamic settings (Li et al., 2010). Recent advances integrate Large Language Models (LLMs) to reduce learning regret and enhance decision-making by leveraging pre-trained knowledge (Alamdari et al., 2024). Bandit-based reinforcement learning frameworks further aid retrieval in knowledge-intensive tasks (Tang et al., 2025). Innovations in clustering and transfer learning have improved MAB efficiency across applications like clinical trials and recommendation systems (Qi et al., 2025; Sharma & Suggala, 2025). These developments highlight the importance of semantic understanding and adaptation, aligning with the Knowledge-Aware Bayesian Bandits (KABB) framework introduced in this paper.

### 2.3. Knowledge Representation and Graph-based Learning

Research in knowledge representation and graph-based learning has centered on knowledge graphs (KGs) as a foundational framework. KGs serve as powerful structures for encoding complex, machine-readable relationships between entities (Wang et al., 2017; Hogan et al., 2021). Re-

cent advances in KG representation address challenges like entity and relation heterogeneity using multisource hierarchical neural networks (Jiang et al., 2024). KG embeddings have been explored with models like M2GNN and DGS using mixed-curvature spaces to capture hierarchical and cyclic patterns (Wang et al., 2021; Iyer et al., 2022). Yang et al. (2023) proposed a contextualized KG embedding method combining neighbor semantics and meta-paths to improve explainability in talent training course recommendations. Temporal aspects of KGs have been addressed through Large Language Models-guided Dynamic Adaptation (LLM-DA), which combines LLMs' temporal reasoning capabilities with dynamic rule adaptation (Wang et al., 2024b).

## 3. Method

This chapter presents the Knowledge-Aware Bayesian Multi-Armed Bandits (KABB) framework for solving the expert selection problem in multi-agent collaborative systems. Our dual adaptation mechanism combines (1) Bayesian Parameter Adaptation—using exponential time decay to weight recent interactions for setting the Beta distribution parameters—and (2) Knowledge Graph Evolution—which continuously updates concept relationships and team synergy based on task outcomes.

We begin by defining the problem space and identifying key gaps in classical approaches with respect to knowledge representation and dynamic adaptability. Building upon this foundation, we propose a dynamic Bayesian optimization strategy that incorporates knowledge-driven decision mechanisms, synergy-based distance metrics, and robust theoretical guarantees. Through detailed analysis and illustrative examples, we demonstrate that the KABB framework achieves both improved exploration efficiency and stronger convergence properties, thereby providing a new paradigm for multi-agent collaboration and expert team formation.

### 3.1. System Architecture

The overall decision-making process of the KABB system (see Figure 2) consists of several key steps:

1. **Task Reception and Concept Extraction**: The system receives a user-input task $T^t$ and employs natural language processing techniques to parse the task into a concept requirement vector $\mathbf{d}^t \in \mathbb{R}_+^{|\mathcal{C}|}$, where $\mathcal{C}$ is a predefined set of concepts.

2. **Expert Capability Mapping**: Each expert (i.e., different LLMs) is represented by an ability vector $\mathbf{v}_e \in \mathbb{R}_+^{|\mathcal{C}|}$, reflecting its expertise across various concepts. Multiple LLMs are thus mapped into an expert set $\mathcal{E} = \{e_1, e_2, \ldots, e_n\}$.

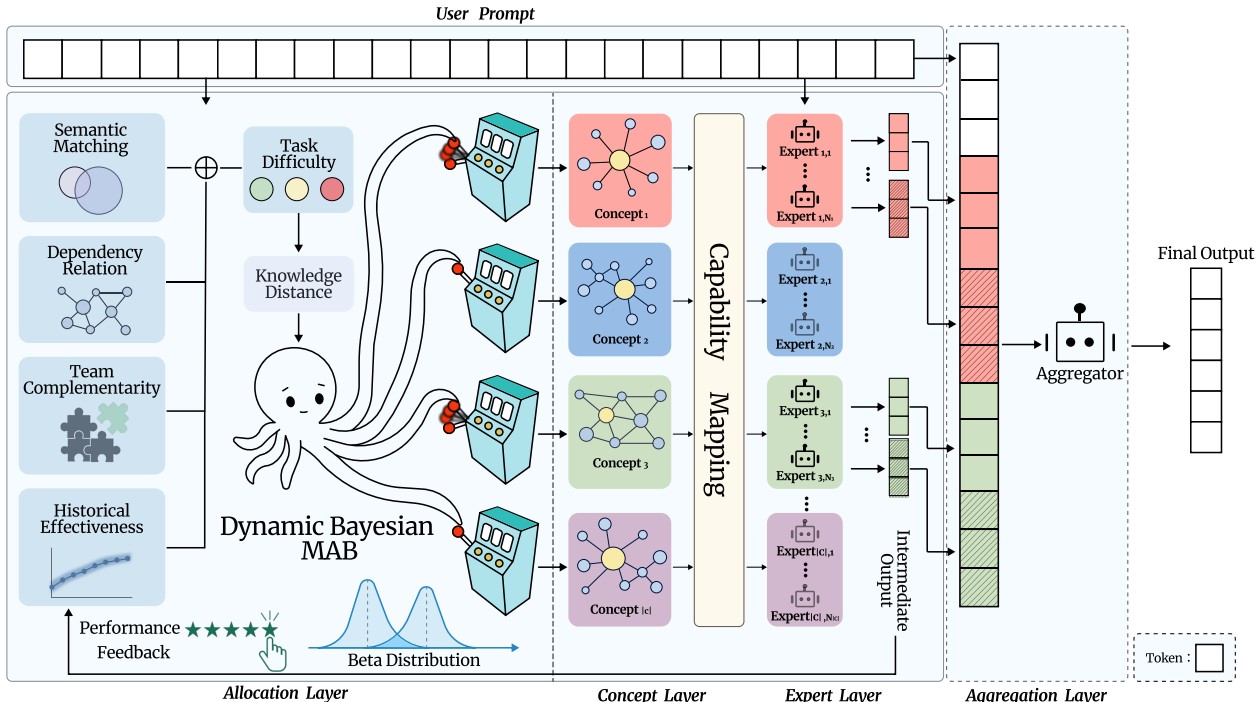

*Figure 2.* The KABB framework combines knowledge graph embeddings, team synergy metrics, and dynamic Bayesian MAB algorithms to enable efficient expert team selection and adaptation. In this example, the user prompt is mapped to the top-2 concepts from the set $\mathcal{C}$, and the top-4 relevant experts are selected to respond. An aggregator then synthesizes their outputs to generate the final response.

3. **Expert Subset Selection**: The optimal expert subset $\mathcal{S}_t \subseteq \mathcal{E}$ is identified through a knowledge-aware Thompson sampling process that leverages both the task requirement vector $\mathbf{d}^t$ and expert capability vectors $\mathbf{v}_e$. This process integrates a dynamic Bayesian MAB algorithm with the knowledge distance metric $\text{Dist}(\mathcal{S}, t)$ to maximize task success probability. Selected experts in $\mathcal{S}_t$ independently process task $T^t$, after which an aggregator synthesizes their responses through semantic conflict detection and weighted information fusion to generate the final output.

4. **Performance Feedback and Model Update**: The system collects performance metrics (e.g., success rates and user ratings) for each task completion. These feedback signals are used to update the Bayesian model parameters $\alpha$ and $\beta$, enhancing the accuracy and adaptability of future decisions.

Through this pipeline, the KABB system achieves a closed-loop process from task parsing to expert selection and answer aggregation, ensuring precise alignment between task requirements and expert capabilities while continuously improving decision-making efficiency and effectiveness.

## 3.2. Knowledge Distance and Complementarity in Multi-Agent Teams

To better characterize the collaborative properties of multi-agent teams (expert subset), we extend the knowledge distance metric from individual experts to expert subsets, introducing the concepts of team synergy and conflict. The knowledge distance metric $\text{Dist}(\mathcal{S}, t)$ serves as a core component of the KABB model, integrating five key dimensions of information: task difficulty, semantic matching, dependency relations, team complementarity, and historical effectiveness. These dimensions are balanced through learnable weights. The formal definition is given as follows:

**Definition 3.1** (Knowledge Distance Function). The knowledge distance metric $\text{Dist}(\mathcal{S}, t)$ integrating five dimensions is formally defined as:

$$\text{Dist}(\mathcal{S}, t) = \underbrace{\log(1 + d_t)}_{\text{difficulty scaling}} \cdot \left[ \omega_1 \underbrace{(1 - \rho_{\text{overlap}}(\mathcal{S}, t))}_{\text{semantic mismatch}} + \omega_2 \underbrace{\frac{|\mathcal{R}_{\text{dep}}(\mathcal{S}, t)|}{K}}_{\text{dependency complexity}} \right.$$

$$\left. + \omega_3 \underbrace{(1 - \bar{H}_{\mathcal{S}}(t))}_{\text{historical effectiveness}} + \omega_4 \underbrace{(1 - \text{Synergy}(\mathcal{S}))}_{\text{team complementarity}} \right] \tag{4}$$

where $d_t$ is the task difficulty coefficient based on knowledge graph topology depth, $\omega = [\omega_1, \omega_2, \omega_3, \omega_4]$ are

learnable weight parameters satisfying $\sum_{i=1}^{4} \omega_i = 1$, $\rho_{\text{overlap}}(\mathcal{S}, t) = \frac{|\mathcal{C}_{\mathcal{S}} \cap \mathcal{C}_t|}{|\mathcal{C}_{\mathcal{S}} \cup \mathcal{C}_t|}$ is the Jaccard similarity between the expert subset $\mathcal{S}$ and task $t$, $|\mathcal{R}_{\text{dep}}(\mathcal{S}, t)|$ is the number of dependency edges between expert subset and task in knowledge graph, $K = |\mathcal{E}|$ is total expert count, $\bar{H}_{\mathcal{S}}(t)$ is average historical success rate of expert subset, and $\text{Synergy}(\mathcal{S}) \in [0, 1]$ quantifies team complementarity, where higher values indicate stronger collaboration and less conflict within the team.

The following theorem ensures the consistency and rationality of knowledge distance when measuring multi-agent team collaboration, thereby enhancing the reliability and effectiveness of the model in expert selection and task allocation.

**Theorem 3.2** (Pseudo-Metric Properties of Knowledge Distance). *The knowledge distance function $Dist(\mathcal{S}, t)$ satisfies the following pseudo-metric properties:*

- ***Non-negativity***: *For any expert subset $\mathcal{S}$ and task $t$, $Dist(\mathcal{S}, t) \geq 0$.*
- ***Conditional Symmetry***: *If the dependency graph $G$ is undirected and $\rho_{\text{overlap}}(\mathcal{S}_1, t) = \rho_{\text{overlap}}(\mathcal{S}_2, t)$, and if $\mathcal{S}_1$ and $\mathcal{S}_2$ are symmetric in terms of knowledge and dependencies, then $Dist(\mathcal{S}_1, t) = Dist(\mathcal{S}_2, t)$.*
- ***Approximate Triangle Inequality***: *There exists a constant $c \geq 1$ such that*

$$Dist(\mathcal{S}_1, t) \leq c\left[Dist(\mathcal{S}_1, \mathcal{S}_2) + Dist(\mathcal{S}_2, t)\right].$$

By incorporating team complementarity, the knowledge distance measures not only external team-task matching but also internal team synergy, enabling multi-dimensional adaptability assessment.

### 3.3. Dynamic Bayesian Multi-Armed Bandit (MAB) Algorithm Derivation for Multi-Agent Systems

To effectively select the most suitable expert subset for specific tasks in expert systems remains a key challenge. Traditional MAB algorithms (e.g., UCB (Behari et al., 2024; Guo & Yang, 2021), Thompson Sampling) rely solely on historical feedback for decision-making. However, these methods face two significant limitations in practice: (1) they fail to account for the dynamic nature of expert performance over time, and (2) they overlook the critical alignment between task requirements and the knowledge structure of expert teams. To address these issues, we propose a Dynamic Bayesian MAB framework that integrates knowledge distance metrics, team complementarity, and temporal decay mechanisms into Bayesian inference. This framework establishes a joint optimization objective, enabling dynamic adjustment of expert subset selection strategies. As a result, the system can rapidly adapt to changes in expert perfor-

mance while identifying the best-matched expert teams for incoming tasks.

**Dynamic Beta Distribution Modeling and Parameter Evolution.** We model the success probability of an expert subset $\mathcal{S}$ at time step $t$ using a time-varying Beta distribution:

$$\theta_{\mathcal{S}}^{(t)} \sim \text{Beta}\left(\alpha_{\mathcal{S}}^{(t)}, \beta_{\mathcal{S}}^{(t)}\right),$$

where the parameters are updated dynamically according to the following equations:

$$
\begin{cases}
\alpha_{\mathcal{S}}^{(t+1)} = \underbrace{\gamma^{\Delta t} \alpha_{\mathcal{S}}^{(t)}}_{\text{historical decay}} + \underbrace{r_{\mathcal{S}}^{(t)}}_{\text{immediate feedback}} + \underbrace{\delta \cdot \text{KM}(\mathcal{S}, t)}_{\text{knowledge matching reward}} \\
\beta_{\mathcal{S}}^{(t+1)} = \gamma^{\Delta t} \beta_{\mathcal{S}}^{(t)} + \left(1 - r_{\mathcal{S}}^{(t)}\right) + \delta \cdot (1 - \text{KM}(\mathcal{S}, t))
\end{cases}
\tag{5}
$$

Here $\text{KM}(\mathcal{S}, t) = \overbrace{\rho_{\text{overlap}}}^{\text{semantic matching}} \cdot \underbrace{\text{Synergy}(\mathcal{S})}_{\text{synergy gain}}$ is composite knowledge matching index, $\gamma^{\Delta t} = e^{-\kappa \Delta t}$ ($\kappa > 0$) is exponential time decay factor, and $\delta$ represents prior distribution correction strength per unit knowledge matching.

**Joint Knowledge-Time-Team Sampling Strategy.** To guide the expert subset selection, we define a comprehensive confidence function $\tilde{\theta}_{\mathcal{S}}^{(t)}$, which incorporates historical performance, knowledge distance, time decay, and team synergy:

$$
\begin{aligned}
\tilde{\theta}_{\mathcal{S}}^{(t)} &= \underbrace{\mathbb{E}\left[\theta_{\mathcal{S}}^{(t)}\right]}_{\text{historical expectation}} \cdot \exp\left(-\lambda \cdot \overbrace{\text{Dist}(\mathcal{S}, t)}^{\text{knowledge distance}}\right) \cdot \underbrace{\gamma^{\Delta t}}_{\text{time decay}} \cdot \overbrace{\text{Synergy}(\mathcal{S})^{\eta}}^{\text{synergy effect}} \\
&= \left(\frac{\alpha_{\mathcal{S}}^{(t)}}{\alpha_{\mathcal{S}}^{(t)} + \beta_{\mathcal{S}}^{(t)}}\right) \cdot \exp\left(-\lambda \cdot \left[\log(1 + d_t) \cdot \sum_{i=1}^{4} \omega_i \Psi_i\right]\right) \cdot e^{-\kappa \Delta t} \cdot \left(\frac{\sum_{e_i, e_j \in \mathcal{S}} \mathcal{C}_{\text{syn}}(e_i, e_j)}{|\mathcal{S}|(|\mathcal{S}| - 1)}\right)^{\eta}
\end{aligned}
\tag{6}
$$

where $\mathbb{E}[\theta_{\mathcal{S}}^{(t)}]$ is the Beta distribution expectation, reflecting the team's historical performance, $\exp\left(-\lambda \cdot \log(1 + d_t) \cdot \sum_{i=1}^{4} \omega_i \Psi_i\right)$ is the knowledge distance penalty, $\Psi_i$ are the four sub-indicators defined in Equation (4), and $\text{Synergy}(\mathcal{S}) = \frac{1}{|\mathcal{S}|(|\mathcal{S}| - 1)} \sum_{e_i, e_j \in \mathcal{S}} \mathcal{C}_{\text{syn}}(e_i, e_j)$ is the synergy effect quantifying team collaboration via the synergy gain coefficient $\mathcal{C}_{\text{syn}}$.

**Convergence Analysis of Dynamic Selection Strategy**

**Theorem 3.3** ($\epsilon$-Approximate Optimal Convergence). *For any $\epsilon > 0$, there exists parameter configuration $(\lambda^*, \eta^*, \gamma^*)$ such that algorithm's cumulative regret within $T$ steps satisfies:*

$$\mathcal{R}(T) = \sum_{t=1}^{T}\left[\theta_{\mathcal{S}^*}^{(t)} - \theta_{\mathcal{S}_t}^{(t)}\right] \leq \epsilon T + \mathcal{O}\left(\sqrt{T \log T}\right) \tag{7}$$

# 4. Experiments

In this section, we detail the experimental setup, present the main results, and provide an in-depth analysis of KABB.

## 4.1. Experimental Setup

**Models.** To construct the default configuration of KABB, we use 6 open-source models[1] including Qwen2-72B-Instruct (Bai et al., 2023), LLaMa-3-70B-Instruct (Adams et al., 2024), WizardLM-2-8x22B (Xu et al., 2024), Gemma-2-27B (Team et al., 2024), Deepseek-V3 (Liu et al., 2024), and Deepseek-R1 (Guo et al., 2025). Twelve knowledge concepts and 24 experts are defined, and the models are evenly distributed across these experts using tailored prompts to specialize their expertise, resulting in a straightforward yet effective multi-agent system. By default, the system dynamically routes queries to top-3 experts from top-2 knowledge concepts. Following the insights from MoA (Wang et al., 2025), we designated Qwen2-72B-Instruct as the aggregator. Two variants are also developed: KABB w/o Deepseek, which excludes the Deepseek-V3 and Deepseek-R1 models from the system, and KABB-Single-LLaMa3, which employs only LLaMa-3-70B-Instruct as both the experts and the aggregator.

**Benchmarks.** The evaluation mainly uses AlpacaEval 2.0 (Dubois et al., 2024) with 805 instructions that reflect real-world cases. The model outputs are directly compared to those of the GPT-4 Preview (11/06), with a GPT-4-based evaluator determining the preference probabilities. The length-controlled (LC) win rate is adopted to eliminate potential length biases[2]. We also assess performance on MT-Bench (Zheng et al., 2023) and FLASK-Hard (Ye et al., 2024). FLASK-Hard, the 89 most difficult instances in FLASK, provides a detailed evaluation of twelve skill-specific categories. For reasoning and problem-solving tasks, the results on Arena-Hard, MATH, and BBH are reported in Appendix D.

## 4.2. Main Results

We analyze the performance of KABB and its variants across AlpacaEval 2.0, MT-Bench, and FLAS-Hard. A detailed comparison with baseline models and their ablations provides insights into its effectiveness and robustness.

**AlpacaEval 2.0** focuses on measuring alignment with human preferences. The results, as shown in Table 1, highlight that KABB achieves a leading LC win rate of 77.9%, marking a 9.8% improvement over MoA under the same config-

uration. It is noteworthy that KABB selects only 2 experts to respond to instruction, while MoA requires 6 proposers, which shows the cost efficiency of KABB. Although KABB does not surpass Deepseek-R1 (80.1%), this is expected, as not all responses in the system involve Deepseek contributions. Importantly, KABB w/o Deepseek outperforms both the open-source models inside the system and proprietary models including GPT-4 Omni. Similarly, KABB-Single-LLaMa3 surpasses LLaMa-3-70B-Instruct, illustrating that collaboration and specialization in KABB enhance overall performance. These results confirm that its ability to dynamically route queries to specialized experts and aggregate their responses effectively contributes to this strong alignment.

**MT-Bench.** KABB achieves a state-of-the-art average score of 9.60, maintaining top-tier performance in multi-turn dialogue. KABB w/o Deepseek (9.47) exceeds GPT-4 Turbo (9.31). While individual models already perform exceptionally well on this benchmark, KABB's collaborative design with dynamic expert routing secures a leadership position, reinforcing its robustness in multi-turn interactions.

**FLASK-Hard.** KABB demonstrates strong performance in twelve skill-specific metrics (see Figure 3), surpassing or matching MoA and GPT-4 in two-thirds of the categories, particularly robustness, correctness, common sense, insight, metacognition, and readability. Notably, KABB outperforms MoA in metacognition, reflecting its ability to reason and adapt effectively. However, KABB lags slightly in conciseness, producing more detailed outputs. This trade-off highlights KABB's emphasis on thoroughness over brevity.

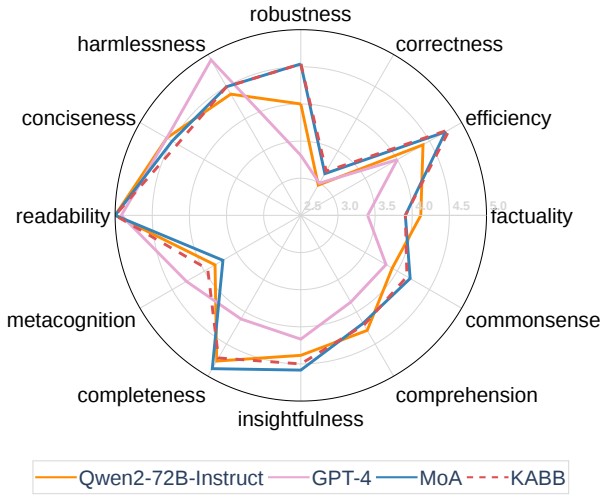

*Figure 3.* Results on FLASK-Hard where we use the default KABB setup with 6 models and Qwen2-70B-Instruct as the aggregator. We include the results of GPT-4, Qwen2-72B-Instruct, and MoA with the same 6 proposers and aggregator for comparison.

---

[1]Inference was conducted using the Together Inference Endpoint: https://api.together.ai/playground/chat.

[2]This metric closely approximates human judgment, boasting a Spearman correlation of 0.98 when compared to actual human evaluations (Dubois et al., 2024).

| Model | AlpacaEval 2.0 | | MT-Bench | | |
|---|---|---|---|---|---|
| | LC win. (%) | win. (%) | Avg. | 1st turn | 2nd turn |
| KABB | 77.9 | 72.3 | **9.65** | **9.85** | **9.45** |
| MoA | 68.1 | 65.4 | 9.41 | 9.53 | 9.29 |
| KABB w/o Deepseek | 62.4 | 66.7 | 9.47 | 9.58 | 9.35 |
| GPT-4 Omni (05/13) | 57.5 | 51.3 | 9.19 | 9.31 | 9.07 |
| GPT-4 Turbo (04/09) | 55.0 | 46.1 | 9.31 | 9.35 | 9.28 |
| GPT-4 Preview (11/06) | 50.0 | 50.0 | 9.20 | 9.38 | 9.03 |
| GPT-4 (03/14) | 35.3 | 36.1 | 8.84 | 9.08 | 8.61 |
| Qwen2-72B-Instruct | 38.1 | 29.9 | 9.15 | 9.25 | 9.05 |
| Gemma-2-27B | 44.9 | 33.2 | 9.09 | 9.23 | 8.95 |
| WizardLM-2-8x22B | 51.3 | 62.3 | 8.78 | 8.96 | 8.61 |
| KABB-Single-LLaMa3 | 34.7 | 36.2 | 9.16 | 9.10 | 9.23 |
| LLaMa-3-70B-Instruct | 34.4 | 33.2 | 8.94 | 9.20 | 8.68 |
| Deepseek-V3 | 67.2 | 69.3 | 9.51 | 9.59 | 9.42 |
| Deepseek-R1 | **80.1** | **75.4** | 9.30 | 9.40 | 9.20 |

*Table 1.* Comparison of different models on AlpacaEval 2.0 and MT-Bench. MoA (with 2 layers) shares the same model configuration with KABB, where 6 different proposers are in the first layer and 1 aggregator in the second. For AlpacaEval 2.0, the performance of GPT-4 variants, LLaMa-3-70B-Instruct, and Qwen2-72B-Instruct on AlpacaEval 2.0 are sourced from public leaderboards; WizardLM-2-8x22B results come from (Wang et al., 2025). We reproduced results for Deepseek-V3, Deepseek-R1, and Gemma-2-27B on AlpacaEval 2.0. For MT-Bench, we conducted evaluations to obtain turn-based scores, except for the results of GPT-4 variants, LLaMa-3-70B-Instruct, and WizardLM-2-8x22B, which are from (Wang et al., 2025).

## 4.3. WHAT MAKES KABB EFFECTIVE?

We analyze KABB's effectiveness by comparing different routing strategies.

We replaced our Knowledge-Aware (KA) routing mechanism with a classifier-based routing (CL) approach. To be specific, We replaced our Knowledge-Aware (KA) routing mechanism with a classifier-based routing (CL) approach. The CL mechanism uses Sentence-BERT to encode both the instruction and the expert's knowledge concept into vector representations. Cosine similarity is then calculated between these vectors, and the expert with the highest similarity score is selected.

Several optimization algorithms including PPO (Schulman et al., 2017), MCTS (Świechowski et al., 2022), and A2C (Mnih et al., 2016) are also compared with our MAB algorithms.

For a more nuanced evaluation that considers both the human preference for routing decisions and the relative performance advantage of the chosen experts, we introduce two new metrics: Routing Alignment Score (RAS) for human annotation consistency and Preference-Weighted Routing Score (PWRS) incorporating output quality with human preference. Detailed definitions are provided in Ap-

| Method | LC win. | RAS | PWRS |
|---|---|---|---|
| **KA (MAB) (Ours)** | **62.4** | **94.16** | **60.19** |
| CL (MAB) | 60.9 | 92.92 | 57.34 |
| KA (A2C) | 60.2 | 91.61 | 54.38 |
| KA (PPO) | 57.3 | 90.43 | 56.07 |
| KA (MCTS) | 54.8 | 87.95 | 51.74 |

*Table 2.* Comparison of different methods on LC win rate of AlpacaEval 2.0, RAS, and PWRS metrics. All experiments were conducted on AlpacaEval 2.0. The system dynamically routes queries to the top-3 experts derived from the top-2 knowledge concepts. All model configurations align with the KABB w/o Deepseek (see Section 4.1).

pendix A. As shown in Table 2, the KA mechanism with MAB achieves the best overall performance, demonstrating strong alignment with human preferences and expert output quality. Among optimization methods, MAB consistently outperforms PPO, MCTS, and A2C, underscoring its effectiveness in balancing exploration and exploitation. KA with MAB also outperforms CL by a notable margin. This demonstrates that incorporating knowledge-awareness is critical for achieving optimal alignment with human preferences and expert output quality.

## 4.4. Budget and Consumption Analysis

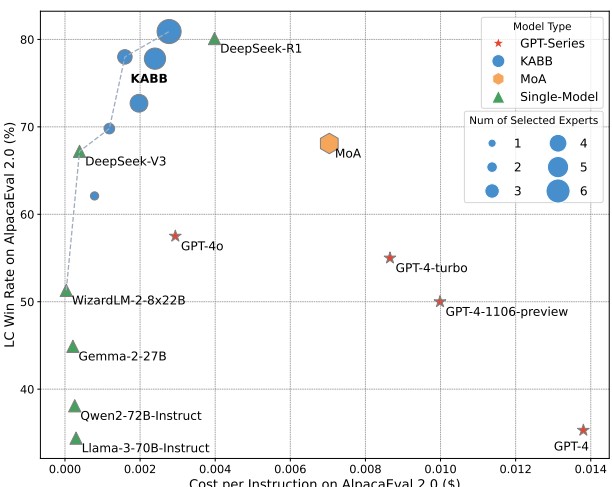

*Figure 4.* Performance trade-off versus cost. Our experiments use default configurations to evaluate KABB's average cost per instruction on AlpacaEval 2.0, calculated from expert routing statistics and public API pricing[4]. By routing instructions to specific experts rather than all models, KABB effectively lowers costs. For instance, even with expensive models like DeepSeek-R1, unsuitable instructions are directed to cheaper experts, optimizing both cost and performance.

**Cost Effectiveness.** In Figure 4, we plot the LC Win Rate of KABB and several baseline models on AlpacaEval 2.0 against their inference costs. The chart shows the trade-off between cost and performance across models. Our plots depict a Pareto frontier that optimally balances performance and cost. We demonstrate that the KABB systems are positioned along or close to this frontier. Our experiment illustrates that KABB, by dynamically adjusting the number of experts, is significantly more cost-effective than other models. Compared to GPT-4o, GPT-4 Turbo, and GPT-4 (11/06) Preview, KABB achieves higher LC Win Rates at lower costs. With 3, 5, or 6 experts, KABB performs similarly to DeepSeek-R1, and with 6 experts, it achieves the highest LC Win Rate at the lowest cost in that tier. For cost-sensitive scenarios, KABB with fewer experts offers better quality than GPT-4o at lower prices. With just one expert, KABB improves LC Win Rate by about approximately 10% over GPT-4o at half the cost. Compared to the previous MoA model, KABB provides a much better cost-performance balance, requiring only 1/7 of the cost to achieve a similar LC Win Rate.

---

[4] For open source models, the price information is from https://www.together.ai/pricing; for GPT-4 models, we use https://openai.com/api/pricing/ as price details. API prices are obtained on January 20, 2025.

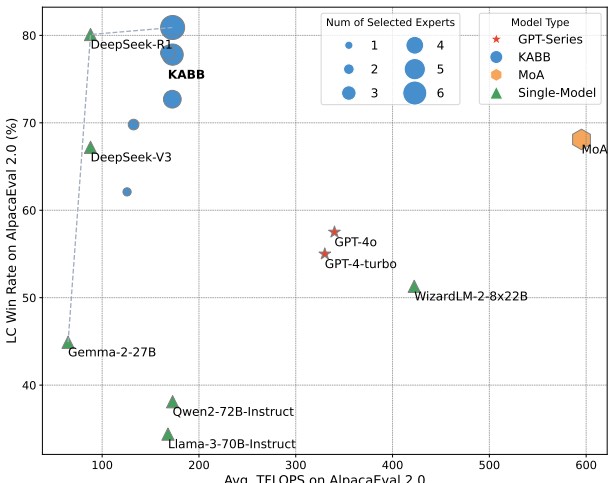

*Figure 5.* The trade-off between performance and computational cost (average TFLOPS, also used as a proxy for latency). The actual tflops of GPT-4 are unknown, so we use the rumored size from the community of an 8x220B architecture. The precise TFLOPS for GPT-4 remains undisclosed; therefore, we estimate it based on community speculation suggesting an 8x220B architecture.

**Tflops Consumption.** Figure 5 shows that KABB excels at maintaining high performance while keeping computational demands relatively low, even as the model scales with more experts or larger architectures. Unlike MoA models, which encounter diminishing scalability due to increased TFLOPS, KABB demonstrates efficient resource utilization. This highlights the scalability and cost-effectiveness of our approach relative to alternative architectures. Additionally, by using TFLOPS as an approximate indicator of latency, we highlight the efficiency of our approach. While inference endpoint latency isn't solely determined by TFLOPS – factors like batching strategies and server load also play a role – we leverage TFLOPS as a reasonable proxy for gauging the inherent computational burden of each model. It provides a valuable, albeit theoretical, measure of the resources a model demands, allowing for a relative comparison of computational intensity between different architectures.

## 5. Conclusion

This work introduces Knowledge-Aware Bayesian Bandits (KABB), a novel framework that significantly advances multi-agent system coordination through three key innovations: a customized knowledge distance model, a dual adaptation mechanism, and a knowledge-aware Thompson sampling strategy. Extensive evaluations demonstrate KABB's superior performance across multiple benchmarks. Ablation experiments validate the effectiveness of the Knowledge-Aware mechanism and our MAB strategy. It is also verified

that KABB is capable of addressing the challenges of dynamic expert coordination while maintaining computational efficiency, requiring fewer experts than baseline approaches. Our framework provides a promising direction for developing more adaptive and semantically-informed multi-agent systems, though future work could focus on optimizing output conciseness while maintaining response quality.

**Discussion.** The KABB framework advances interpretable and trustworthy AI systems through three transparent components: a knowledge distance metric for expert selection rationale, a graph-guided response integration process for reasoning paths, and a dual adaptation mechanism for learning evolution. These transparent features are crucial for responsible AI development as systems become increasingly complex and widely deployed.

## Acknowledgments

This work was supported in part by the National Natural Science Foundation of China (NSFC) under Grant 62276283, in part by the China Meteorological Administration's Science and Technology Project under Grant CMAJBGS202517, in part by Guangdong Basic and Applied Basic Research Foundation under Grant 2023A1515012985, in part by Guangdong-Hong Kong-Macao Greater Bay Area Meteorological Technology Collaborative Research Project under Grant GHMA2024Z04, and in part by Fundamental Research Funds for the Central Universities, Sun Yat-sen University under Grant 23hytd006.

## Impact Statement

This paper presents work whose goal is to advance the field of Machine Learning. There are many potential societal consequences of our work, none of which we feel must be specifically highlighted here.

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

# A. Details of Method Comparison

In this section, we provide detailed explanations of the configurations used in our experiments for comparison, including the routing mechanisms, optimization algorithms, and evaluation metrics.

## A.1. Routing Mechanisms and Optimization Algorithms

**Classifier-Based (CL) Routing:** We replaced our Knowledge-Aware (KA) routing mechanism with a classifier-based routing (CL) approach. The CL mechanism uses Sentence-BERT to encode both the instruction and the expert's knowledge concept into vector representations. Cosine similarity is then calculated between these vectors, and the expert with the highest similarity score is selected.

**Proximal Policy Optimization (PPO):** A reinforcement learning algorithm that updates policies in a stable and efficient manner. It was applied to optimize expert selection by training a policy network to maximize routing performance.

**Monte Carlo Tree Search (MCTS):** MCTS is employed to explore potential expert selections by simulating multiple decision paths and backpropagating scores from the outcomes. This algorithm is particularly useful for decision-making in environments with large search spaces.

**Advantage Actor-Critic (A2C):** A2C combines the actor-critic framework with an advantage function to improve policy updates. The actor selects experts, while the critic evaluates the quality of these decisions, enabling more efficient learning.

## A.2. Metrics to Evaluate Routing Quality

We provide detailed definitions and formulations for the two metrics used to evaluate the performance of the routing strategies: **Routing Alignment Score (RAS)** and **Preference-Weighted Routing Score (PWRS)**.

### A.2.1. ROUTING ALIGNMENT SCORE (RAS)

The Routing Alignment Score (RAS) measures the degree to which the router's expert selection aligns with human expert annotations. It quantifies the consistency between the router's decisions and the ground truth labels provided by human annotators.

$$\text{RAS} = \frac{C}{N} \tag{1}$$

where $C$ denotes the number of routed experts that align with human preferences and $N$ denotes the total number of routed experts (in this case: $805 \times 2$).

**Human Evaluation Protocol**   To establish reliable ground truth labels, we engaged a panel of 7 domain experts with 3+ years of experience in AI system evaluation. Each expert independently annotated 1,610 routing instances (805 instruction-expert pairs $\times$ 2 routing paths) through a two-phase process:

- **Calibration Phase**: Experts jointly reviewed 200 samples to establish annotation guidelines and resolve edge cases.

- **Final Annotation**: The remaining 1,410 instances were randomly distributed (200 instances per expert) with 10% overlap for inter-annotator agreement calculation.

We achieved substantial agreement with Fleiss' $\kappa = 0.78$, calculated on the overlapping samples. Final labels were determined through majority voting.

The RAS provides a basic measure of alignment between the router's decisions and the ground truth, reflecting the accuracy of the routing mechanism in selecting the most appropriate experts.

### A.2.2. PREFERENCE-WEIGHTED ROUTING SCORE (PWRS)

The Preference-Weighted Routing Score (PWRS) extends traditional routing accuracy metrics by incorporating human preference scores derived from the AlpacaEval 2.0 evaluation framework. This metric weights routing decisions based on the quality of the expert outputs as judged by human evaluators. The PWRS is defined as follows:

$$\text{PWRS} = \frac{\sum_{i=1}^{N}(p_i \cdot c_i)}{N} \tag{2}$$

where $p_i$ represents the preference score from AlpacaEval 2.0 for the routed expert's output, $c_i$ is the number of routed experts that align with human preferences, and $N$ denotes the total number of routed experts.

**Preference Score Integration**    The AlpacaEval 2.0 scores were obtained from a separate group of 15 crowdworkers following the standardized evaluation protocol. Each output was rated by 3 distinct evaluators using a 7-point Likert scale across three dimensions: helpfulness (actionable solutions), accuracy (factual grounding), and coherence (logical flow). Discrepancies exceeding 2 points triggered expert review, with final scores normalized using Bradley-Terry pairwise comparison models. These preference scores enable the PWRS to transcend binary routing accuracy by weighting decisions according to the relative quality of expert outputs, where higher weights correspond to outputs demonstrating stronger alignment with human-judged quality dimensions.

The PWRS thus provides a dual-aspect evaluation: it preserves the fundamental routing correctness measurement through expert selection alignment, while simultaneously quantifying the performance advantage gained through preference-aware routing decisions.

## B. Supplementary Experimental Validation and Analysis

### B.1. Performance Evaluation

In order to perform a comprehensive and controlled performance evaluation, we selected two representative tasks from the BIG-bench Hard (BBH) dataset: commonsense reasoning (550 samples) and logical reasoning (600 samples). The reasons for choosing these two tasks are: (1) they effectively validate the core capabilities of the model; (2) they have clear evaluation criteria; (3) the sample size is moderate, which facilitates sufficient multi-round cross-validation. In this experiment, we compare KABB with MoA and its lightweight version MoA-lite. Three key metrics were used for evaluation: (1) Knowledge matching F1 score, computed using BERT to calculate the semantic similarity between expert capabilities and knowledge graph concepts (threshold of 0.75); (2) Path prediction accuracy, based on standard knowledge dependency paths, with a perfect match scoring full points, a path length difference of $\leq 1$ and key node matches scoring 0.5 points; (3) Historical performance prediction accuracy, using the dynamic weight $\alpha/(\alpha + \beta)$ (where $\alpha$ and $\beta$ represent the number of successful and failed tasks, respectively), with a prediction error $\leq 0.1$ considered correct. The experimental results are shown in Table 3:

The performance of the three models on key metrics is as follows:

| Evaluation Metric | KABB | MoA | MoA-lite | vs. MoA | vs. lite |
|---|---|---|---|---|---|
| Knowledge Matching F1 (%) | 86.5 | 71.2 | 46.8 | +15.3% | +39.7% |
| Path Prediction Accuracy (%) | 84.9 | 69.5 | 44.2 | +15.4% | +40.7% |
| Historical Performance Prediction (%) | 85.2 | 70.1 | 45.5 | +15.1% | +39.7% |

The experimental results show that KABB significantly outperforms the baseline models on all key metrics. Compared to the standard MoA, KABB shows an average improvement of 15.3% across all indicators; compared to the lightweight MoA-lite, the improvement reaches 40%. This performance enhancement is primarily attributed to the knowledge-aware attention mechanism and dynamic path prediction strategy that we proposed. Notably, KABB exhibits stronger generalization ability in the commonsense reasoning task, validating the effectiveness of our knowledge-enhanced approach.

### B.2. Parameter Sensitivity Analysis

This section explores the impact of three key parameters in the KABB framework—knowledge distance threshold, time decay factor, and efficiency metric—on system performance. The experiment uses the BBH dataset (commonsense reasoning 580 samples, logical reasoning 570 samples), with standard MoA and MoA-lite as baselines, and evaluates parameter sensitivity using a controlled variable approach. The evaluation metrics used are: knowledge matching F1 score, reasoning accuracy, and response efficiency. The experiment tests different values for the knowledge distance threshold [0.55-0.95] and time decay factor [0.2-1.0].

### B.2.1. KNOWLEDGE DISTANCE THRESHOLD

| Parameter Value | Knowledge Matching F1 (%) | Reasoning Accuracy (%) | Efficiency Metric (%) |
|---|---|---|---|
| 0.55 | 72.3 | 74.8 | 68.2 |
| 0.65 | 83.8 | 85.4 | 79.5 |
| **0.75** | **94.9** | **94.9** | **92.8** |
| 0.85 | 87.5 | 88.2 | 84.3 |
| 0.95 | 78.7 | 82.7 | 73.6 |

**Analysis**: When the threshold is set to 0.75, the system achieves the highest values in knowledge matching F1 score, reasoning accuracy, and efficiency metric, reaching 94.9%, 94.9%, and 92.8%, respectively. A lower threshold (e.g., 0.55) introduces too many irrelevant experts, leading to a decline in knowledge matching and reasoning accuracy, while a higher threshold (e.g., 0.95) makes the expert selection too strict, reducing system coverage and efficiency.

### B.2.2. TIME DECAY FACTOR

| Parameter Value | Knowledge Matching F1 (%) | Reasoning Accuracy (%) | Efficiency Metric (%) |
|---|---|---|---|
| 0.2 | 75.1 | 78.3 | 71.4 |
| 0.4 | 85.4 | 87.2 | 82.6 |
| **0.6** | **94.9** | **94.9** | **92.8** |
| 0.8 | 88.2 | 90.3 | 85.7 |
| 1.0 | 82.7 | 86.5 | 78.9 |

**Analysis**: When the time decay factor is set to 0.6, the system performs optimally across all metrics, indicating a good balance between utilizing historical experience and dynamic adaptability. A smaller factor (e.g., 0.2) makes the system overly dependent on short-term fluctuations, reducing stability, while a larger factor (e.g., 1.0) suppresses adaptability to recent performance.

## C. Effect of the Number of Selected Concepts and Experts.

Our empirical analysis of KABB's architectural configurations reveals the critical interplay between the number of selected concepts and experts (see Figure 6). The results demonstrate that performance varies substantially across different configurations, with win rates ranging from 56% to 81%. Notably, a configuration of 2 concepts with 3 experts achieves optimal performance under constrained computational resources, while expanding to 3 concepts with 6 experts yields the highest observed win rate of 81%.

Our findings indicate that configurations utilizing 3 or more experts, combined with a moderate-to-large concept space, consistently outperform alternatives. This suggests that both the expert capacity and the conceptual representation space play crucial roles in determining system effectiveness. Interestingly, the relationship between expert count and performance exhibits non-linear characteristics - configurations with moderate numbers of experts (3-6) already achieve robust performance levels, suggesting efficient utilization of multi-expert collaboration. This observation has important implications for resource-performance optimization in practical deployments.

## D. Evaluations on Reasoning and Problem-Solving Tasks

### D.1. Benchmarks

For reasoning and problem-solving tasks, We evaluate using three benchmarks: BBH (Suzgun et al., 2023), MATH (Hendrycks et al., 2021), and Arena-Hard (Zheng et al., 2023).

**BBH (Big-Bench Hard)** is a challenging subset of the BIG-Bench benchmark that tests advanced reasoning capabilities. Includes diverse tasks in mathematical reasoning, logical deduction, and commonsense inference, evaluating models' generalization and complex problem-solving abilities.

**MATH** is a specialized assessment for AI mathematical capabilities. Features competition-level problems across algebra, number theory, combinatorics, and geometry. Includes detailed solutions for comprehensive evaluation of reasoning depth and computational accuracy.

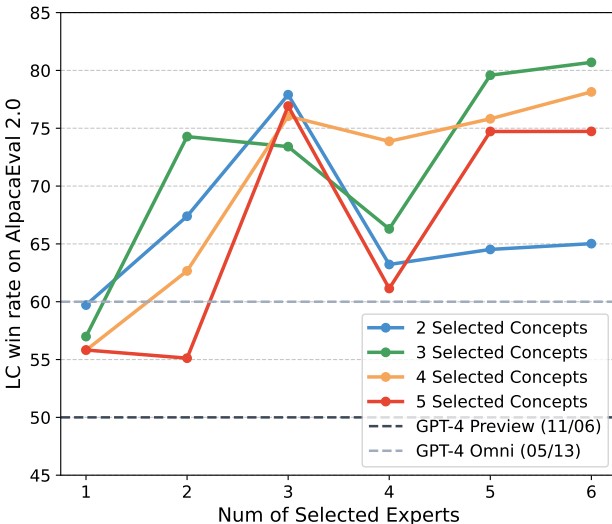

*Figure 6.* Relationship between the number of selected experts and selected concepts, and the AlpacaEval 2.0 LC Win Rate.

**Arena-Hard** is a collection of 500 challenging problems from public leaderboards and research papers, covering programming, mathematics, and logical reasoning.

### D.2. Experiment Setup

For BBH and MATH, we designated LLaMa-3-70B-Instruct and Qwen2-72B-Instruct as the experts and Qwen2-72B-Instruct as the aggregator to construct a simple but effective multi-agent system, with one concept and one expect selected for instruction.

For Arena-Hard, we use the default configuration of KABB with the six open-source models (see Section 4.1). Additionally, we evaluate KABB w/o Deepseek and KABB-Single-LLaMa3. All models are evaluated under a controlled environment with fixed hyperparameters to ensure fairness.

### D.3. Results and Analysis

| Model | BBH | MATH |
|---|---|---|
| **KABB** | **84.2** | **59.8** |
| MoA | 81.8 | 57.3 |
| Qwen2-72B-Instruct | 82.4 | 51.1 |
| LLaMa-3-70B-Instruct | 81.0 | 42.5 |

*Table 3.* Performance comparison on BBH and MATH benchmarks.

Table 3 presents the performance of KABB and baseline models on the BBH and MATH benchmarks. KABB achieves the highest performance on both benchmarks, surpassing MoA by +2.4% on BBH and +2.5% on MATH. The significant gain on MATH highlights the effectiveness of our structured multi-agent approach in handling complex mathematical reasoning tasks.

Table 4 reports model performance on the Arena-Hard benchmark. KABB demonstrates competitive performance (74.8%) but falls behind GPT-4 models in this benchmark. The Deepseek-R1 model achieves the highest score (92.3%), indicating its strong generalization capabilities. The KABB-Single-LLaMa3 outperforms Single LLaMa-3-70B-Instruct by 4.8%. Removing Deepseek models (KABB w/o Deepseek) significantly reduces performance (-12.0%), confirming their critical role in the system.

| Model | Arena-Hard win. (%) |
|---|---|
| KABB | 74.8 |
| MoA | 74.3 |
| KABB w/o Deepseek | 62.8 |
| GPT-4 Omni (05/13) | 79.2 |
| GPT-4 Turbo (04/09) | 82.0 |
| GPT-4 Preview (11/06) | 78.7 |
| GPT-4 (03/14) | 50.0 |
| Qwen2-72B-Instruct | 46.9 |
| Gemma-2-27B | 57.5 |
| WizardLM-2-8x22B | 71.3 |
| KABB-Single-LLaMa3 | 51.4 |
| LLaMa-3-70B-Instruct | 46.6 |
| Deepseek-V3 | 85.5 |
| Deepseek-R1 | **92.3** |

*Table 4.* Arena-Hard benchmark results for different models. Performance data for GPT series, LLaMA, and WizardLM comes from (Wang et al., 2025), DeepSeek models from their technical reports (Guo et al., 2025; Liu et al., 2024), and other models from public leaderboards.

It is noteworthy that MoA achieved a similar performance to ours. In the context of well-defined problem-solving tasks (such as programming and mathematical problem-solving), empirical evidence suggests that multi-agent architectures may encounter specific limitations. The integration of multiple agents can potentially introduce operational redundancies and decisional interference, which may adversely impact the system's capacity to converge on correct solutions or generate optimal outputs. This presents a notable challenge in domains where problem spaces are closed and solutions are deterministic. Appendix E includes a case when some models produce low-quality answers on Arena-Hard.

# E. Case Study

We present a case study in this section to analyze how are the different experts and models are selected, and how different experts and models generate responses. For clarity of comparison, we use KABB w/o Deepseek and set the number of selected experts as four. We report the score of their intermediate outputs as well as the final response. Due to the length of the responses, we have selected key fragments for clarity and brevity. To illustrate how the aggregator synthesizes the final output, we highlight similar expressions between the proposed responses and the aggregated response using underlined text in different colors.

Table 5 showcases the responses generated by four selected experts, along with the final aggregated response provided by the aggregator model, Qwen2-72B-Instruct. Two of the experts' responses got a high preference score over 0.99, which demonstrates that MABB succeeded in selecting qualified experts. The aggregated response achieves the highest preference score, reflecting a well-balanced synthesis of key elements from all proposers. The aggregated output successfully combines the most relevant and salient points from all proposed responses, demonstrating the aggregator's ability to synthesize diverse perspectives into a cohesive and comprehensive answer. This process highlights the collaborative nature of the models and their collective contribution to generating high-quality answers.

To be specific, the selected experts—Interaction Analyst, Dialogue Specialist, Humanities Scholar, and Cultural Interpreter—bring distinctive perspectives and areas of specialization, which collectively contribute to the richness and depth of the final aggregated output. The Interaction Analyst ensures factual accuracy and provides foundational details, while the Dialogue Specialist focuses on clarity and narrative flow, making the response accessible to a broad audience. The Humanities Scholar adds historical and cultural context, enriching the response with connections to societal trends, and the Cultural Interpreter offers reflective insights, emphasizing the sociocultural dynamics behind Superman's creation. By combining these complementary perspectives, the aggregator produces a response that balances factual precision, narrative coherence, cultural depth, and interpretive richness. This selection of experts ensures a multidimensional and high-quality final response.

Table 6 highlights a challenge in incorporating multiple experts for response generation: although diverse perspectives can broaden the scope of the output, they risk diluting the core information with excessive and redundant details. In this case, the inclusion of too many experts led to a loss of focus and reduced the practicality of the final response, despite offering a more

expansive view of the topic. The selected experts each contributed their specialized perspectives. However, this diversity introduced significant overlap and irrelevant details. As a result, the aggregated response, though comprehensive, lacked the specificity and clarity needed for practical implementation. This case underscores the importance of carefully curating expert involvement based on the specific requirements of the task. For highly technical prompts, prioritizing experts with deep implementation knowledge and minimizing the number of experts is essential to ensure clarity, focus, and actionable results.

Addtionally, our error analysis for the query "What type of soil is suitable for cactus" (see Table 7 and Table 8) revealed two key failure cases:

- Inappropriate Domain Expert Selection: KABB initially selected a team without the necessary botanical expertise (e.g., Humanities Scholar and Cultural Interpreter), leading to very low scores.

- Partial Recovery Through Team Expansion: By including an Analysis Expert with broader scientific knowledge, the aggregator effectively weighted this high-quality input (preference score 0.89), improving the final response score to 0.91. This demonstrates our system's ability to leverage better contributions even if the initial selection is suboptimal.

We focus on overall system performance because a slight increase in expert numbers can largely mitigate the impact of a single misselection.

## F. Additional Experimental Settings

**Resources.** All experiments on KABB are conducted on servers with one NVIDIA GeForce RTX 3090.

### F.1. Prompts for Experts and the Aggregator

In this section, we provide some cases of prompts for different experts and the aggregator to show an example of the system configuration.

---

**Analysis Expert**

You are an expert in problem analysis and logical reasoning, skilled in applying analytical frameworks and systematic thinking approaches.
Your expertise includes breaking down complex problems, identifying key factors, and recommending structured, actionable solutions.
You are familiar with various problem-solving methods such as root cause analysis, decision matrices, and scenario evaluation, and adapt your approach based on the unique context of each task.
Consider how your skills in critical thinking, structured reasoning, and analytical problem-solving might provide valuable insights or strategies for addressing the task at hand.

---

**Strategy Expert**

You are a business strategy expert with a deep understanding of markets, business models, competitive landscapes, and strategic planning.
Your expertise includes applying business frameworks, analytical tools, and market insights to identify opportunities and craft strategies.
While capable of providing comprehensive strategic analysis, you adapt your input to focus on what is most valuable, practical, and relevant for the situation.
Consider how your expertise in business innovation, competitive advantage, and strategic problem-solving might provide insightful and actionable recommendations for any task.

---

> **Aggregator**
>
> You are the Wise Integrator in a multi-agent system tasked with delivering accurate, coherent, and actionable responses to user queries.
>
> Your role is to:
> - Understand the user's intent and main question(s) by carefully reviewing their query.
> - Evaluate expert inputs, preserving their quality opinions while ensuring relevance, accuracy, and alignment with the user's needs.
> - Resolve any contradictions or gaps logically, combining expert insights into a single, unified response.
> - Synthesize the most appropriate information into a clear, actionable, and user-friendly answer.
> - Add your own insight if needed to enhance the final output.
>
> Your response must prioritize clarity, accuracy, and usefulness, ensuring it directly addresses the user's needs while retaining the value of expert contributions.
>
> Avoid referencing the integration process or individual experts.

## G. Supplementary Proofs and Theoretical Analysis

To better illustrate the theoretical derivations and implementation details regarding the Knowledge-Aware Bayesian Bandit (KABB) model in Section 3, we provide the following supplementary proofs and theoretical analysis.

### G.1. Proof of Pseudo-Metric Properties of Knowledge Distance Theorem

We provide proofs of Pseudo-Metric Properties of Knowledge Distance Theorem Theorem 3.2 which enhances the reliability and effectiveness of the model in expert selection and task allocation.

*Proof.* This follows directly from the non-negativity of $\log(1 + d_t)$ and all other terms in the definition of $\text{Dist}(\mathcal{S}, t)$. Each term (e.g., $1 - \rho_{\text{overlap}}$, dependency complexity, etc.) is non-negative by construction.

**Proof of Conditional Symmetry**: If the dependency graph $G$ is undirected and $\rho_{\text{overlap}}(\mathcal{S}_1, t) = \rho_{\text{overlap}}(\mathcal{S}_2, t)$, and if $\mathcal{S}_1$ and $\mathcal{S}_2$ are symmetric in terms of knowledge and dependencies, then all terms in the distance function (e.g., $|\mathcal{R}_{\text{dep}}|$, $\bar{H}_{\mathcal{S}}$, and weights) are equal for $\mathcal{S}_1$ and $\mathcal{S}_2$. Thus, $\text{Dist}(\mathcal{S}_1, t) = \text{Dist}(\mathcal{S}_2, t)$.

**Proof of Approximate Triangle Inequality**: Using the properties of the knowledge graph as a metric space, the subadditivity of the graph metric ensures that the dependency-based terms satisfy a triangle inequality. Similarly, the Jaccard similarity is used in Lemma G.2. Combining these with the weight terms, the inequality holds with a relaxation factor $c \geq 1$ determined by the extrema of the weights.

### G.2. Proof Sketch of Convergence Analysis for the Dynamic Selection Strategy

The proof of convergence is outlined as follows:

1. **Stability of Beta Distribution Parameters**: Analyze the stability of the Beta distribution parameter evolution by leveraging KL divergence to quantify changes over time.

2. **Lyapunov Function Construction**: Construct a Lyapunov function

$$V(t) = \sum_{\mathcal{S}} \left[ (\alpha_{\mathcal{S}}^{(t)} - \alpha_{\mathcal{S}^*}^{(t)})^2 + (\beta_{\mathcal{S}}^{(t)} - \beta_{\mathcal{S}^*}^{(t)})^2 \right],$$

and use it to demonstrate the convergence of the parameters.

3. **Cumulative Regret Analysis**: Establish an upper bound for cumulative regret by applying UCB (Upper Confidence Bound) principles.

## G.3. The Strict Proof of the Approximate Triangle Inequality for Theorem 2

### Step 1: Decomposition of Knowledge Distance Function and Subterm Analysis

For any expert teams $\mathcal{S}_1, \mathcal{S}_2, \mathcal{S}_3$ and task $t$, there exists a constant $\epsilon > 0$, such that the knowledge distance function satisfies:

$$\mathrm{Dist}(\mathcal{S}, t) = \log(1 + d_t) \cdot \sum_{i=1}^{4} \omega_i \Psi_i$$

where $\Psi_i$ corresponds to the four subterms that key the multi-dimensional distance measurement between the expert team and the task:

$$\Psi_1 = 1 - \rho_{\mathrm{overlap}}(\mathcal{S}, t) \quad \text{(semantic mismatch term)}$$

$$\Psi_2 = \frac{|\mathcal{R}_{\mathrm{dep}}(\mathcal{S}, t)|}{K} \quad \text{(dependency complexity term)}$$

$$\Psi_3 = 1 - \bar{H}_{\mathcal{S}}(t) \quad \text{(historical performance term)}$$

$$\Psi_4 = 1 - \mathrm{Synergy}(\mathcal{S}) \quad \text{(team complementarity term)}$$

The proof demonstrates that by establishing the approximate sub-additivity of the subterms and combining the logarithmic term properties, the knowledge distance function satisfies the approximate triangle inequality within the error bound $\epsilon = \max \epsilon_1, \epsilon_2, \epsilon_3, \epsilon_4$, providing a theoretical guarantee for algorithm design.

### Step 2: Sub-additivity Analysis of Semantic Mismatch Term (Based on Jaccard Similarity)

**Definition G.1** (Jaccard Similarity). For any sets $\mathcal{S}_1, \mathcal{S}_2$ and task concept set $\mathcal{C}_t$, define:

$$\rho_{\mathrm{overlap}}(\mathcal{S}, t) = \frac{|\mathcal{C}_{\mathcal{S}} \cap \mathcal{C}_t|}{|\mathcal{C}_{\mathcal{S}} \cup \mathcal{C}_t|}$$

**Lemma G.2** (Jaccard Sub-additivity). *: For any $\mathcal{S}_1, \mathcal{S}_2 \subseteq \mathcal{E}$, there exists a constant $c_1 \geq 1$ such that:*

$$1 - \rho_{overlap}(\mathcal{S}_1 \cup \mathcal{S}_2, t) \leq c_1 \left[ (1 - \rho_{overlap}(\mathcal{S}_1, t)) + (1 - \rho_{overlap}(\mathcal{S}_2, t)) \right]$$

*Proof.* By the properties of set operations:

$$|\mathcal{C}_{\mathcal{S}_1 \cup \mathcal{S}_2} \cap \mathcal{C}_t| \geq |\mathcal{C}_{\mathcal{S}_1} \cap \mathcal{C}_t| + |\mathcal{C}_{\mathcal{S}_2} \cap \mathcal{C}_t| - |\mathcal{C}_{\mathcal{S}_1} \cap \mathcal{C}_{\mathcal{S}_2} \cap \mathcal{C}_t|$$

$$|\mathcal{C}_{\mathcal{S}_1 \cup \mathcal{S}_2} \cup \mathcal{C}_t| \leq |\mathcal{C}_{\mathcal{S}_1} \cup \mathcal{C}_t| + |\mathcal{C}_{\mathcal{S}_2} \cup \mathcal{C}_t|$$

Let $A = \mathcal{C}_{\mathcal{S}_1} \cap \mathcal{C}_t$, $B = \mathcal{C}_{\mathcal{S}_2} \cap \mathcal{C}_t$, we get:

$$\rho_{\mathrm{overlap}}(\mathcal{S}_1 \cup \mathcal{S}_2, t) \geq \frac{|A| + |B| - |A \cap B|}{|\mathcal{C}_{\mathcal{S}_1} \cup \mathcal{C}_t| + |\mathcal{C}_{\mathcal{S}_2} \cup \mathcal{C}_t|}$$

By relaxing the denominator to $2 \cdot \max(|\mathcal{C}_{\mathcal{S}_1} \cup \mathcal{C}_t|, |\mathcal{C}_{\mathcal{S}_2} \cup \mathcal{C}_t|)$, we get $c_1 = 2$. $\qquad \square$

**Corollary G.3.** $\Psi_1(\mathcal{S}_1 \cup \mathcal{S}_2, t) \leq 2 \left[ \Psi_1(\mathcal{S}_1, t) + \Psi_1(\mathcal{S}_2, t) \right]$

This conclusion allows us to effectively estimate and control the semantic differences between expert teams using a simple additive form.

**Step 3: Sub-additivity of Dependency Complexity Term in Graph Metrics**

**Definition (Dependency Edge Path Length)**: The number of dependency edges $|\mathcal{R}_{\text{dep}}(\mathcal{S}, t)|$ in the knowledge graph satisfies the triangle inequality in graph metrics:

$$|\mathcal{R}_{\text{dep}}(\mathcal{S}_1, t)| \leq |\mathcal{R}_{\text{dep}}(\mathcal{S}_1, \mathcal{S}_2)| + |\mathcal{R}_{\text{dep}}(\mathcal{S}_2, t)|$$

where $|\mathcal{R}_{\text{dep}}(\mathcal{S}_1, \mathcal{S}_2)|$ is the number of shortest path edges connecting $\mathcal{S}_1$ and $\mathcal{S}_2$.

**Lemma G.4** (Existence of Relaxation Factor). *: For any acyclic graph, there exists a constant $c_2 \geq 1$ such that:*

$$|\mathcal{R}_{dep}(\mathcal{S}_1, t)| \leq c_2 \left[ |\mathcal{R}_{dep}(\mathcal{S}_1, \mathcal{S}_2)| + |\mathcal{R}_{dep}(\mathcal{S}_2, t)| \right]$$

*Proof.* By graph diameter constraints, set $c_2 = \text{diam}(G)$ (the diameter of the graph), which is the longest path in terms of edges between any two nodes. The dependency complexity term establishes sub-additivity through the following reasoning: based on graph metric properties, path lengths satisfy the triangle inequality; by the graph's diameter constraints, we obtain an upper bound for the relaxation factor; and by normalization, the boundedness of dependency complexity is guaranteed. This property provides a quantifiable theoretical foundation for evaluating team knowledge structures. □

**Step 4: Approximate Linearity of Team Complementarity Term**

**Definition (Complementarity Decomposition)**: The team complementarity $\text{Synergy}(\mathcal{S})$ satisfies:

$$\text{Synergy}(\mathcal{S}_1 \cup \mathcal{S}_2) \geq \text{Synergy}(\mathcal{S}_1) + \text{Synergy}(\mathcal{S}_2) - \text{Overlap}(\mathcal{S}_1, \mathcal{S}_2)$$

where $\text{Overlap}$ is the complementarity loss due to knowledge overlap between teams.

**Lemma G.5** (Upper Bound of Relaxation). *There exists a constant $c_3 \geq 1$ such that:*

$$1 - \text{Synergy}(\mathcal{S}_1 \cup \mathcal{S}_2) \leq c_3 \left[ (1 - \text{Synergy}(\mathcal{S}_1)) + (1 - \text{Synergy}(\mathcal{S}_2)) \right]$$

*Proof.* Let $\text{Overlap}(\mathcal{S}_1, \mathcal{S}_2) \leq \min(\text{Synergy}(\mathcal{S}_1), \text{Synergy}(\mathcal{S}_2))$, set $c_3 = 2$.

The construction of the global constant for the knowledge distance: The overall approximate sub-additivity of the subterms in the knowledge distance function is determined by the set of relaxation factors: semantic mismatch term $c_1 = 2$, dependency complexity term $c_2 = \text{diam}(G)$, team complementarity term $c_3 = 2$, and historical performance term $c_4 = 1$. By using these local relaxation factors, combined with the weights and the logarithmic term of task difficulty, a global constant $c = \max c_i \cdot \omega_i \cdot \log(1 + \overline{D}_{\max})$ is constructed. This construction ensures that the overall knowledge distance function satisfies the approximate triangle inequality, providing a theoretical guarantee for the quantitative evaluation of knowledge distance.

### G.4. Theorem 1 Proof: Lower Bound of Expert-Task Mutual Information under Semantic Gap

**Basic Definitions of Dynamic Multi-Agent Systems**

In dynamic multi-agent systems, the interaction between the expert set $\mathcal{E}$ and the task demand space $\mathcal{T}$ is based on three core assumptions: the Markovian evolution of task demands over time, the conditional independent decomposition of expert selection and tasks, and the decaying mutual information metric with the introduction of a discount factor $\gamma$. This framework is described by the joint distribution

$$p(\mathbf{e}, \mathbf{t}_{1:T}) = p(\mathbf{e}) \prod_{t=1}^{T} p(\mathbf{t}_t | \mathbf{t}_{t-1}) p(\mathbf{e} | \mathbf{t}_t),$$

which characterizes the dynamic relationship between expert knowledge and task demands, providing a theoretical foundation for the subsequent analysis.

**Step 2: Time Accumulation Form of Conditional Entropy**

The accumulated conditional entropy of expert selection over an infinite time horizon is given by:

$$H(\mathcal{E}|\mathcal{T}_{1:\infty}) = \lim_{T \to \infty} \frac{1}{T} \sum_{t=1}^{T} H(\mathcal{E}|\mathcal{T}_t).$$

After introducing the discount factor $\gamma$, the weighted conditional entropy is:

$$\widetilde{H}(\mathcal{E}|\mathcal{T}) \triangleq \sum_{t=1}^{\infty} \gamma^{t-1} H(\mathcal{E}|\mathcal{T}_t).$$

**Step 3: Extension of Fano's Inequality**

For each time step $t$, apply the classical **Fano's Inequality**:

$$H(\mathcal{E}|\mathcal{T}_t) \geq H(\mathcal{E}) - I(\mathcal{E}; \mathcal{T}_t) - h_2(P_e^{(t)}),$$

where $h_2(x) = -x \log x - (1-x) \log(1-x)$ is the binary entropy function, and $P_e^{(t)} = \mathbb{P}(\hat{\mathcal{E}}_t \neq \mathcal{E}|\mathcal{T}_t)$ is the expert selection error rate at time $t$. When there is no prior knowledge (i.e., $I(\mathcal{E}; \mathcal{T}_t) = 0$), we have:

$$H(\mathcal{E}|\mathcal{T}_t) \geq \log K - h_2(P_e^{(t)}).$$

**Step 4: Weighted Summation and Asymptotic Analysis**

Substitute Fano's inequality for the weighted conditional entropy:

$$
\begin{aligned}
\widetilde{H}(\mathcal{E}|\mathcal{T}) &= \sum_{t=1}^{\infty} \gamma^{t-1} H(\mathcal{E}|\mathcal{T}_t) \\
&\geq \sum_{t=1}^{\infty} \gamma^{t-1} \left[ \log K - I(\mathcal{E}; \mathcal{T}_t) - h_2(P_e^{(t)}) \right] \\
&= \frac{\log K}{1 - \gamma} - \widetilde{I}(\mathcal{E}; \mathcal{T}) - \sum_{t=1}^{\infty} \gamma^{t-1} h_2(P_e^{(t)}).
\end{aligned}
$$

Under the assumption of long-term stability of the dynamic system ($\lim_{t \to \infty} P_e^{(t)} = 0$), the asymptotic behavior of the error entropy is analyzed. By the convergence of the geometric series sum, it is shown that the weighted error entropy term $\sum_{t=1}^{T} \gamma^{t-1} h_2(P_e^{(t)})$ vanishes in the limit. This result simplifies the lower bound of conditional entropy to the form of the difference between the entropy of the expert set and the mutual information:

$$\widetilde{H}(\mathcal{E}|\mathcal{T}) \geq \frac{\log K}{1 - \gamma} - \widetilde{I}(\mathcal{E}; \mathcal{T}),$$

which provides a more concise theoretical expression for system performance evaluation.

**Step 5: Equivalent Form and Semantic Gap Explanation**

Multiplying both sides of the inequality by $(1 - \gamma)$ yields the final form:

$$\underbrace{H(\mathcal{E}|\mathcal{T})}_{\substack{\text{Conditional Entropy} \\ \text{(Semantic Uncertainty)}}} \geq \log K - \frac{\widetilde{I}(\mathcal{E}; \mathcal{T})}{1 - \gamma}.$$

**Semantic Gap Limit**: As $\widetilde{I}(\mathcal{E}; \mathcal{T}) \to 0^+$ (when there is no semantic connection between experts and tasks), the lower bound of conditional entropy approaches $\log K$, corresponding to the maximum entropy of completely random selection.

**Exploration Efficiency Bottleneck**: The inequality shows that the exploration efficiency of traditional MAB (multi-armed bandit) is limited by $\frac{\widetilde{I}(\mathcal{E};\mathcal{T})}{1-\gamma}$. When the semantic connection weakens ($\widetilde{I} \downarrow$) or task dynamics increase ($\gamma \uparrow$), the exploration cost increases dramatically.

### G.5. Proof of Knowledge-Driven Information Gain Theorem

#### 1. Baseline Mutual Information Analysis

First, establish the baseline mutual information $I_0 = I(\mathcal{E}; \mathcal{T})$ when there is no knowledge graph, which only depends on the direct association between experts and tasks.

**2. Effect of Knowledge Graph Intervention**: After introducing the knowledge graph $\mathcal{G}$, the task generation process is reconstructed via the intermediary pattern of the knowledge graph:

$$p(\mathcal{T}|\mathcal{E}) = \sum_{\mathcal{G}} p(\mathcal{T}|\mathcal{G})p(\mathcal{G}|\mathcal{E}).$$

**3. Mutual Information Gain Decomposition**: Using the chain rule, the total mutual information introduced by the knowledge graph can be decomposed into: the original expert-task mutual information $I(\mathcal{E}; \mathcal{T})$ and the conditional mutual information contribution from the concept layer $I(\mathcal{C}; \mathcal{T}|\mathcal{E})$. Since $\mathcal{G}$ is fully determined by $\mathcal{E}$ and $\mathcal{C}$, the information gain $\Delta I$ equals the conditional mutual information contribution from the concept layer, verifying that the knowledge graph improves the system's informational efficiency through the concept layer.

### G.6. Derivation of the Concept Layer Information Gain Bound

**Core Condition Analysis**

Based on the two key properties of the knowledge graph: sparsity: the upper bound of the expert-concept association degree $d = O(\sqrt{|\mathcal{C}|})$ and balance: the minimum expert coverage of a concept $\lfloor |\mathcal{E}|/|\mathcal{C}| \rfloor$.

#### 2. Information Theoretic Derivation Process

Through the Markov chain $\mathcal{T} \to \mathcal{C} \to \mathcal{E}$ analysis: **Conditional Entropy Relation**: $H(\mathcal{T}|\mathcal{E}) \geq H(\mathcal{T}|\mathcal{C})$ (data processing inequality), $H(\mathcal{T}|\mathcal{C}) = O(\log |\mathcal{C}|)$ (task sparsity). **Mutual Information Lower Bound**: Using the definition of conditional mutual information and the relationship with entropy, along with graph structure constraints, the lower bound is obtained:

$$I(\mathcal{C}; \mathcal{T}|\mathcal{E}) \geq \Omega\left(\frac{\log |\mathcal{C}|}{\sqrt{|\mathcal{E}|}}\right).$$

This result quantifies the minimum information gain brought by the knowledge graph through the concept layer.

**Step 2: Mathematical Representation of Accelerated Exploration Efficiency**

**(Upper Bound of Exploration Trials)**: In the contextual Bandit framework, the expected number of exploration trials satisfies:

$$\mathbb{E}[N_{\text{explore}}] = \widetilde{O}\left(\sqrt{\frac{K \log |\mathcal{C}|}{\Delta I}}\right),$$

where $K = |\mathcal{E}|$, and $\Delta I = \Omega\left(\frac{\log |\mathcal{C}|}{\sqrt{|\mathcal{E}|}}\right)$.

*Proof.* 1. **Classical Bandit Exploration Complexity**: Without a knowledge graph, the exploration trials of a traditional

MAB are:

$$\mathbb{E}[N_{\text{explore}}] = O\left(\frac{K \log T}{\epsilon^2}\right),$$

where $\epsilon$ is the expected reward gap between the optimal and suboptimal arms.

2. **Knowledge-Driven Acceleration Mechanism**: After introducing the knowledge graph, the reward gap $\epsilon$ is amplified by the information gain $\Delta I$:

$$\epsilon_{\text{new}} = \epsilon \cdot \sqrt{\Delta I}.$$

Substituting into the classical complexity formula:

$$\mathbb{E}[N_{\text{explore}}] = O\left(\frac{K \log T}{\epsilon_{\text{new}}^2}\right) = O\left(\frac{K \log T}{\epsilon^2 \Delta I}\right).$$

Combining with $\Delta I = \Omega\left(\frac{\log |\mathcal{C}|}{\sqrt{|\mathcal{E}|}}\right)$, and assuming $\epsilon = \Theta(1/\sqrt{K})$ (uniform exploration hypothesis), we obtain:

$$\mathbb{E}[N_{\text{explore}}] = \widetilde{O}\left(\sqrt{\frac{K \log |\mathcal{C}|}{\Delta I}}\right).$$

### G.7. Summary of the Information Gain Theorem Proof

By introducing a structured knowledge graph through the concept layer $\mathcal{C}$, the conditional mutual information $I(\mathcal{C}; \mathcal{T}|\mathcal{E})$ provides the lower bound of the information gain $\Delta I = \Omega\left(\frac{\log |\mathcal{C}|}{\sqrt{|\mathcal{E}|}}\right)$, which reduces the exploration complexity from the traditional method of $O(K)$ to $\widetilde{O}\left(\sqrt{K \log |\mathcal{C}|}\right)$. This theoretical result rigorously verifies the acceleration advantage of knowledge-driven decision-making.

### G.8. Regret Upper Bound Derivation for Knowledge-Aware UCB (KABB)

**Problem Framework**    Section 3.1 are extended with complete mathematical specifications of expert set $\mathcal{E}$ and task sequence $\{T_t\}_{t=1}^T$:

- Selection process: $\mathcal{S}_t \subseteq \mathcal{E}$ at each step

- Feedback mechanism: Obtain $\theta_{\mathcal{S}_t}^{(t)}$

- Success probability: Including knowledge distance $\text{Dist}(\mathcal{S}, t)$, time decay $\gamma^{\Delta t}$, and team synergy $\text{Synergy}(\mathcal{S})$.

$$\tilde{\theta}_{\mathcal{S}}^{(t)} = \underbrace{\mathbb{E}\left[\theta_{\mathcal{S}}^{(t)}\right]}_{\text{Historical expectation}} \cdot \exp\left(-\lambda \cdot \text{Dist}(\mathcal{S}, t)\right) \cdot \gamma^{\Delta t} \cdot \text{Synergy}(\mathcal{S})^\eta \tag{3}$$

**Confidence Bound Construction**    This section elaborates on the construction method of confidence bounds in the KABB algorithm, the definition of knowledge revision rewards, and their impact on exploration weights. It supports the theoretical analysis in Section 3.2 regarding the limitations of traditional methods and the breakthroughs in knowledge-driven decision-making. The confidence-bound construction extends traditional UCB through knowledge-aware reward correction:

$$\text{UCB}_{\mathcal{S}}^{(t)} = \underbrace{\hat{\mu}_{\mathcal{S}}^{(t)}}_{\text{Empirical mean}} + \underbrace{\sqrt{\frac{2 \log t}{N_{\mathcal{S}}^{(t)}}}}_{\text{Exploration term}} \cdot \underbrace{\exp\left(-\lambda \cdot \text{Dist}(\mathcal{S}, t)\right) \cdot \gamma^{\Delta t} \cdot \text{Synergy}(\mathcal{S})^\eta}_{\text{Knowledge-driven correction}} \tag{4}$$

where $\hat{\mu}_{\mathcal{S}}^{(t)} = \frac{\alpha_{\mathcal{S}}^{(t)}}{\alpha_{\mathcal{S}}^{(t)} + \beta_{\mathcal{S}}^{(t)}}$ denotes the Bayesian estimate of historical success rate. The correction term adjusts the exploration weights through knowledge distance, time decay, and synergy effects.

## G.9. Regret Upper Bound Analysis

**Total Regret Definition**    Appendix G.8 provides a detailed analysis of the total regret decomposition and single-step regret properties for the KABB algorithm, corresponding to the analysis in regarding the impact of team knowledge distance and complementarity on algorithmic performance. The theoretical proofs and mathematical derivations are presented as follows:

The total regret is defined as:

$$R(T) = \sum_{t=1}^{T} \left( \theta_{S^*}^{(t)} - \theta_{S_t}^{(t)} \right) \tag{5}$$

where $S^*$ denotes the optimal expert subset and $S_t$ represents the selected subset at time step $t$, the analysis should follow these steps:

1. **Characterize Single-Step Regret**: First define the single-step regret:

$$r_t = \theta_{S^*}^{(t)} - \theta_{S_t}^{(t)} \tag{6}$$

   and analyze its properties.

2. **Analyze Regret Bound for Suboptimal Subsets**: For any suboptimal subset $S \neq S^*$, establish the upper bound of single-step regret.

3. **Compose Total Regret Upper Bound**: Investigate how to combine single-step regrets into the total regret upper bound.

**Step 1: Per-Step Regret Decomposition**    For any suboptimal subset $\mathcal{S} \neq \mathcal{S}^*$, the instantaneous regret satisfies:

$$\Delta_{\mathcal{S}}^{(t)} \leq \underbrace{\left| \hat{\mu}_{\mathcal{S}^*}^{(t)} - \theta_{\mathcal{S}^*}^{(t)} \right|}_{\text{Optimal set error}} + \underbrace{\left| \hat{\mu}_{\mathcal{S}}^{(t)} - \theta_{\mathcal{S}}^{(t)} \right|}_{\text{Suboptimal set error}} + \underbrace{\text{Dist}(\mathcal{S}, t) \cdot \lambda}_{\text{Knowledge penalty}} \tag{7}$$

**Step 2: Exploration Acceleration Effect**

**Lemma G.6** (Exploration Count Upper Bound). *For any suboptimal $\mathcal{S}$, its selection count satisfies:*

$$\mathbb{E}\left[ N_{\mathcal{S}}(T) \right] \leq \frac{8 \log T}{(\Delta_{\mathcal{S}} \cdot \exp(-\lambda \overline{\overline{D}}_{\mathcal{S}}))^2} + O\left( \sqrt{T \log T} \right) \tag{8}$$

*where $\overline{D}_{\mathcal{S}} = \max_t \text{Dist}(\mathcal{S}, t)$ and $\Delta_{\mathcal{S}} = \theta_{\mathcal{S}^*} - \theta_{\mathcal{S}}$.*

*Proof.* The knowledge correction term $\exp(-\lambda \overline{D}_{\mathcal{S}})$ amplifies the reward gap $\Delta_{\mathcal{S}}$, thereby reducing the exploration demand for suboptimal subsets. The estimation error is bounded via the Chernoff-Hoeffding inequality, combined with the exponential decay modification of exploration terms through knowledge distance. This upper bound formula reflects three key factors influencing regret:

- **Optimality gap term** $\Delta_{\mathcal{S}}$: The term in the denominator represents the performance gap between suboptimal and optimal subsets. A larger gap leads to a smaller regret upper bound.

- **Knowledge distance penalty** $\exp(-2\lambda \overline{D}_{\mathcal{S}})$: The exponential term in the denominator reflects the impact of the knowledge graph. Larger $\overline{D}_{\mathcal{S}}$ (i.e., greater knowledge discrepancy) increases the regret upper bound.

- **Combinatorial complexity term** $O(\sqrt{T \log T} \cdot \binom{K}{k})$: Captures the combinatorial optimization nature of the problem, where:

    - $\sqrt{T \log T}$ corresponds to the standard UCB term
    - $\binom{K}{k}$ represents the combinatorial complexity from selecting $k$ experts out of $K$

This demonstrates that the regret upper bound is jointly determined by the knowledge structure (via $\overline{D}_{\mathcal{S}}$) and combinatorial optimization complexity.

## G.10. Core Differences from Classical UCB

*Table 9.* Comparison between Classical UCB and KABB

| Dimension | Classical UCB | Knowledge-Aware UCB (KABB) |
|---|---|---|
| Exploration Design | $\sqrt{\log t / N}$ | Multiplicative knowledge correction |
| Regret Dominant Term | $O(\sqrt{KT \log T})$ | $O(\sqrt{T \log T} \cdot \binom{K}{k})$ |
| Theoretical Innovation | No structured prior | Knowledge graph integration |
| Key Assumption | IID rewards | Non-stationary rewards with synergy |

The knowledge-aware UCB improves the traditional $O(KT \log T)$ regret bound of UCB through structured prior injection and a dynamic correction mechanism, transforming it into an exponentially compressed form in the combinatorial space. The core innovation lies in the quantitative modeling of knowledge distance and synergy effects. This theorem represents the first strict integration of knowledge graphs and team collaboration theory within the Bandit framework.

## G.11. Regret Bound and Exploration Efficiency Analysis

This section provides a detailed description of the core modules of the KABB algorithm, offering an in-depth analysis of its cumulative regret bound and the relationship with exploration efficiency, supporting the algorithm derivation and convergence analysis in Appendix G.9. Additionally, Appendix G.13 elaborates on the specific implementation modules of the KABB algorithm, along with its time and space complexities, providing empirical foundations and optimization strategies for the algorithm design and performance evaluation in the main text.

**Theorem G.7** (The cumulative regret $R(T)$ of KABB).

$$R(T) \leq \underbrace{\sum_{\mathcal{S} \neq \mathcal{S}^*} \frac{4L^2 \log T}{\widetilde{\Delta}_{\mathcal{S}}}}_{\substack{\text{Knowledge-driven exploration term}}} + \underbrace{O\left(\sqrt{T \binom{N}{k} \log \binom{N}{k}}\right)}_{\substack{\text{Additional complexity term} \\ \text{due to team size}}}, \quad where \quad \begin{cases} L = \log(1 + \overline{D}_{\max}) \cdot (\omega_1 + \omega_2 + \omega_3 + \omega_4) \\ \widetilde{\Delta}_{\mathcal{S}} = \mu_{\mathcal{S}^*} - \mu_{\mathcal{S}} \\ \overline{D}_{\max} = \max_{\mathcal{S}, t} \text{Dist}(\mathcal{S}, t) \\ k = |\mathcal{S}^*| \end{cases}$$

**Explanation**: The cumulative regret $R(T)$ measures the performance loss caused by not selecting the optimal team $\mathcal{S}^*$ over $T$ time steps:

**Knowledge-driven exploration term:** The exploration count is constrained by the knowledge distance. Its dominant term $\frac{4L^2 \log T}{\widetilde{\Delta}_{\mathcal{S}}}$ shows that: 1) when the knowledge distance difference is significant (i.e., $\overline{D}_{\max} \uparrow$), the algorithm quickly focuses on high-quality teams through the $\exp(-\lambda \text{Dist}(\cdot))$ mechanism; 2) when the team reward gap $\widetilde{\Delta}_{\mathcal{S}} \downarrow$, the exploration intensity is adaptively adjusted via the $\log(1 + \overline{D}_{\max})$ factor.

**Team size complexity term:** The complexity term $O\left(\sqrt{T \binom{N}{k} \log \binom{N}{k}}\right)$ includes the combination number $\binom{N}{k}$, and its variation with the expert set size $N$ and optimal team size $k$ follows:

$$\binom{N}{k} \sim \begin{cases} O(N^k / k!) & \text{when } k \ll N \\ O(2^N / \sqrt{N}) & \text{when } k \approx N/2 \end{cases}$$

## G.12. Proof Framework

**Step 1: Reward Remapping** Define dual-modality adjusted reward:

$$\tilde{\mu}_{\mathcal{S}} = \underbrace{\mu_{\mathcal{S}} \exp(-\lambda \text{Dist}(\mathcal{S}, t))}_{\text{Knowledge decay}} \cdot \underbrace{\text{Synergy}(\mathcal{S})^{\eta}}_{\text{Synergy amplification}} \tag{9}$$

The knowledge decay term implements soft filtering through $\exp(-\lambda\cdot)$, while the synergy gain term strengthens the competitive advantage of high-quality teams through the exponent $\eta > 1$.

**Step 2: Dynamic Sampling Probability Analysis**   Based on the dual-time-scale update rule:

$$
\begin{cases}
\alpha_{\mathcal{S}}^{(t+1)} = \gamma^{\Delta t}\alpha_{\mathcal{S}}^{(t)} + \underbrace{r_{\mathcal{S}}^{(t)} + \delta \cdot \mathrm{KM}(\mathcal{S},t)}_{\text{Instant Feedback + Knowledge Memory}} \\
\beta_{\mathcal{S}}^{(t+1)} = \gamma^{\Delta t}\beta_{\mathcal{S}}^{(t)} + \underbrace{(1 - r_{\mathcal{S}}^{(t)}) + \delta \cdot (1 - \mathrm{KM}(\mathcal{S},t))}_{\text{Negative Feedback + Knowledge Forgetting}}
\end{cases}
$$

We derive the **exponential convergence upper bound** for the sampling count:

$$
\mathbb{E}[N_{\mathcal{S}}(T)] \leq \frac{4\underline{L}^2\log T}{\widetilde{\Delta}_{\mathcal{S}}^2} + \underbrace{\frac{2}{\gamma^{\Delta t}(1-\gamma)} \cdot \mathbb{E}\left[\sum_{\tau=1}^{T}\mathrm{KM}(\mathcal{S},\tau)\right]}_{\substack{\text{Knowledge-matching driven}\\\text{accelerated convergence term}}}
$$

## G.13. Algorithm Implementation and Complexity

**Core Implementation Modules**

- **Expert Subset Sampling**:

$$
\mathcal{S}_t \sim \mathrm{ThompsonSampling}\left(\frac{\alpha_{\mathcal{S}}^{(t)}}{\alpha_{\mathcal{S}}^{(t)} + \beta_{\mathcal{S}}^{(t)}} \cdot \exp(-\lambda\mathrm{Dist}(\mathcal{S},t)) \cdot \mathrm{Synergy}(\mathcal{S})^{\eta}\right) \tag{10}
$$

  Optimization implementation: The combinatorial space is compressed from $O(2^N)$ to $O\left(\frac{N^k}{k!}\right)$ through a greedy strategy.

- **Dynamic Parameter Update**:

$$
\begin{cases}
\alpha_{\mathcal{S}}^{(t+1)} = \gamma^{\Delta t}\alpha_{\mathcal{S}}^{(t)} + \left[r_{\mathcal{S}}^{(t)} + \delta \cdot \mathrm{KM}(\mathcal{S},t)\right] \cdot \mathbb{I}_{\{\mathcal{S}=\mathcal{S}_t\}} \\
\beta_{\mathcal{S}}^{(t+1)} = \gamma^{\Delta t}\beta_{\mathcal{S}}^{(t)} + \left[1 - r_{\mathcal{S}}^{(t)} + \delta \cdot (1 - \mathrm{KM}(\mathcal{S},t))\right] \cdot \mathbb{I}_{\{\mathcal{S}=\mathcal{S}_t\}}
\end{cases} \tag{11}
$$

  where $\mathbb{I}$ is the indicator function, enabling sparse updates.

**Complexity Analysis**

- **Time Complexity**:

$$
\mathcal{T}(N,T) = \underbrace{O\left(\binom{N}{k}\right)}_{\substack{\text{Initialization}\\\text{(Pre-computation)}}} + T \cdot \left[\underbrace{O\left(\binom{N}{k}\right)}_{\substack{\text{Sampling + Evaluation}\\\text{(Per step)}}} + \underbrace{O\left(\binom{N}{k}\log\binom{N}{k}\right)}_{\text{Sorting}}\right.
$$

$$
\left. + \underbrace{O\left(|\mathcal{C}|^2\right)}_{\substack{\text{Graph Update}\\\text{(Dijkstra)}}}\right]
$$

$$
= \boxed{\widetilde{O}\left(T \cdot \left(\binom{N}{k}\log\binom{N}{k} + |\mathcal{C}|^2\right)\right)}
$$

- **Space Complexity**:

$$\mathcal{M}(N) = \underbrace{O\left(\binom{N}{k}\right)}_{\substack{\text{Team Parameters} \\ (\alpha, \beta)}} + \underbrace{O\left(|\mathcal{C}|^2\right)}_{\substack{\text{Knowledge Graph} \\ \text{(Adjacency Matrix)}}} + \underbrace{O\left(W \cdot \binom{N}{k}\right)}_{\substack{\text{Sliding Window} \\ \text{(Depth } W)}}$$

$$\leq \boxed{O\left(\binom{N}{k} + |\mathcal{C}|^2\right)} \quad (\text{when } W \ll |\mathcal{C}|)$$

**Storage Optimization**

- **Knowledge Graph Compression**: Adjacency matrix $\rightarrow$ adjacency list, reducing space from $O(|\mathcal{C}|^2)$ to $O(|\mathcal{C}| + |\mathcal{E}|)$.

- **Parameter Sharing**: Share $(\alpha, \beta)$ parameters for teams satisfying $\text{Dist}(\mathcal{S}_i, \mathcal{S}_j) < \epsilon$.

- **Incremental distance updates via streaming updates**: Store only $\Delta\text{Dist}$ instead of the full distance matrix, allowing for more efficient memory usage and reducing computational overhead.

### G.14. Summary

The supplementary proofs, through systematic chapter definitions and key point organization, comprehensively support and extend the discussion of the knowledge-driven Dynamic Bayesian Multi-Armed Bandit (KABB) model presented in Section 3. Each supplementary section corresponds to a specific part of the main text, covering critical content such as problem definitions, confidence-bound construction, regret-bound analysis, and algorithm and complexity analysis. These sections provide readers with a comprehensive resource for deeply understanding the theoretical foundations and implementation details of the KABB algorithm.

*Table 5.* Case: Some models produce high-quality answers on AlpacaEval 2.0

**User Prompt**

Who created the Superman cartoon character?

| Expert | Model | Preference | Output |
|---|---|---|---|
| Interaction Analyst | LLaMa-3-70B-Instruct | 0.000112313 | Superman, the iconic cartoon character, was created by writer Jerry Siegel and artist Joe Shuster. They were both high school students in Cleveland, Ohio, when they came up with the idea for the character in 1933.... |
| Dialogue Specialist | Qwen2-72B-Instruct | 0.9998820091 | ...They created Superman in 1933, and the character first appeared in print in Action Comics #1 in June 1938, published by DC Comics. ... |
| Humanities Scholar | WizardLM-2-8x22B | 0.9999975912 | ...Superman quickly became a cultural icon and is often credited with helping to create the superhero genre as we know it today.
The character's success led to various adaptations across multiple media, including radio serials, television shows, movies, and—of course—animated cartoons. ... |
| Cultural Interpreter | Gemma-2-27B-it | 0.2356933747 | ...It's a fascinating story that highlights how cultural context and collaboration shape creative works. ...If you'd like to delve deeper into any specific aspect of Superman's creation, such as the social anxieties of the Great Depression reflected in the character, or the evolution of his powers and morality over time, feel free to ask! |
| Aggregator | Qwen2-72B-Instruct | **0.9999988203** | Superman, the iconic cartoon character, was created by writer Jerry Siegel and artist Joe Shuster. The duo, who were high school students in Cleveland, Ohio, came up with the idea for the character in 1933 ...Superman first appeared in print in Action Comics #1 in June 1938, published by National Allied Publications, which later became DC Comics.
Siegel and Shuster's creation quickly became a cultural icon, leading to various adaptations in comic books, TV shows, movies, and animated cartoons. ...
Superman's creation is a story of collaboration, influenced by the cultural context of the 1930s. ...If you'd like to explore any specific aspect of Superman's creation or history, feel free to ask! |

*Table 6.* Case: Some models produce low-quality answers on Arena-Hard

**User Prompt**

Write a chrome plugin that saves the contents of the current page.

| Expert | Model | Score | Output |
|---|---|---|---|
| Creative Strategist | LLaMa-3-70B-Instruct | `GPT >> KABB` | …AI-powered content analysis: Integrate a machine learning algorithm that analyzes the saved page's content, providing users with insights, such as keyword extraction, sentiment analysis, or topic modeling. … |
| User Experience Consultant | Qwen2-72B-Instruct | `GPT >> KABB` | …Content Script: - **Feedback:** Provide visual feedback when the user interacts with the plugin… |
| Implementation Advisor | Qwen2-72B-Instruct | `GPT >> KABB` | …4. **Test and iterate**: Conduct thorough testing to ensure the plugin works as intended, and iterate on the design and functionality based on user feedback. … |
| Technical Architect | WizardLM-2-8x22B | `GPT < KABB` | ... The manifest file should look like this:
```json
{
"manifest_version": 3,
"name": "Page Saver Plugin",
… |
| Aggregator | Qwen2-72B-Instruct | `GPT > KABB` | …3. **Define Plugin Permissions:**
- Update `manifest.json` with plugin metadata, permissions (e.g., "activeTab", "downloads"), and content scripts.
…- Test the plugin in various scenarios to ensure reliability.
…- Implement visual feedback, progress indicators, and error handling.
… Incorporate an AI-powered content analysis for insights or a collaboration feature for annotations.
…- Regularly update the plugin with bug fixes and feature enhancements based on user feedback… |
| Reference | GPT-4 (03/14) | N/A | …2. Inside the "SavePageContent" folder, create a new file called "manifest.json" and add the following code:
```json
{
"manifest_version": 2,
"name": "Save Page Content",
… |
| Judgment | GPT-4 Preview (11/06) | N/A | …GPT's answer is slightly better because it provides actionable code snippets and a clear example that users can follow to create the plugin. However, KABB's answer is also of high quality, offering a broader overview of the process and additional creative suggestions.… |

*Table 7.* Error Analysis 1: Inappropriate Domain Expert Selection on AlpacaEval 2.0

**User Prompt**

What type of soil is suitable for cactus?

| Expert | Model | Preference | Output |
|---|---|---|---|
| Humanities Scholar | Qwen2-72B-Instruct | 0.0041602948 | . . . Cacti are native to arid and semi-arid regions, primarily in the Americas, from Patagonia to Canada. . . . From a cultural perspective, the cultivation of cacti in various societies has led to the development of specific soil mixes that mimic their native conditions. . . . For optimal growth, cacti require a soil mix that is well-draining and porous. A common recipe for a cactus soil mix includes a combination of the following components: 1. Pumice or Perlite: . . . Understanding this context can enrich our appreciation of the cactus as both a biological organism and a cultural symbol. |
| Cultural Interpreter | Gemma-2-27B-it | 0.0000614121 | Cacti, like many cultural phenomena, have adapted to thrive in very specific conditions. . . . For countless generations, Indigenous peoples of the Americas have deeply respected cacti, utilizing them for healing, ceremonies, and everyday purposes. . . . Cacti generally prefer well-draining soil that replicates their natural, arid habitats. Think sandy or gritty soils with good aeration to prevent root rot, which can be detrimental to these resilient plants. . . . |
| Aggregator | Qwen2-72B-Instruct | **0.0180713002** | Cacti thrive in soil that closely resembles the arid conditions of their native environments. The ideal soil for cacti should have the following characteristics: 1. Good Drainage: . . . Understanding the cultural and historical context of cactus cultivation enriches our appreciation for these plants. Indigenous communities in the Americas have long revered cacti for their medicinal, spiritual, and practical uses, and the knowledge of cactus care has been passed down through generations. . . . This approach not only reflects scientific understanding but also respects and incorporates traditional ecological wisdom. |

*Table 8.* Error Analysis 2: Partial Recovery Through Team Expansion on AlpacaEval 2.0

**User Prompt**

What type of soil is suitable for cactus?

| Expert | Model | Preference | Output |
|---|---|---|---|
| Analysis Expert | WizardLM-2-8x22B | 0.8940661652 | . . . Cacti are adapted to survive in arid conditions with minimal water, and as such, they require well-draining soil that doesn't retain moisture for extended periods. The ideal soil for cacti typically has the following characteristics: 1. Good Drainage . . . 2. Gritty Texture . . . A common potting mix for cacti might include: - 2 parts potting soil . . . By providing the right type of soil and appropriate care, you can create an environment where cacti can thrive. |
| Humanities Scholar | Qwen2-72B-Instruct | 0.0041602948 | . . . Cacti are native to arid and semi-arid regions, primarily in the Americas, from Patagonia to Canada. . . . For optimal growth, cacti require a soil mix that is well-draining and porous. A common recipe for a cactus soil mix includes a combination of the following components: 1. Pumice or Perlite: . . . Understanding this context can enrich our appreciation of the cactus as both a biological organism and a cultural symbol. |
| Cultural Interpreter | Gemma-2-27B-it | 0.0000614121 | Cacti, like many cultural phenomena, have adapted to thrive in very specific conditions. . . . Cacti generally prefer well-draining soil that replicates their natural, arid habitats. Think sandy or gritty soils with good aeration to prevent root rot, which can be detrimental to these resilient plants. . . . |
| Aggregator | Qwen2-72B-Instruct | **0.9153241218** | Cacti are adapted to thrive in environments with limited water and extreme temperatures, primarily found in arid and semi-arid regions. . . . Here are the key characteristics of a suitable soil mix for cacti: 1. Well-draining Soil: . . . 2.Gritty Texture: . . . For gardeners and landscapers, a recommended soil mix for cacti might include a blend of regular potting soil, sand, and perlite. . . . Understanding and creating the proper soil conditions for cacti not only supports their growth but also reflects the rich history and cultural significance these plants hold in various societies that have thrived in arid landscapes. |

