# OpenReview forum: "KABB: Knowledge-Aware Bayesian Bandits for Dynamic Expert Coordination in Multi-Agent Systems"
_ICML.cc/2025/Conference — ICML 2025 poster_

### Official Review · Reviewer_22fq · 2025-02-23

**Overall Recommendation:** 3

**Summary:**

This paper introduces Knowledge-Aware Bayesian Bandits (KABB) as a novel framework for improving dynamic expert coordination in multi-agent systems, which defines a five-dimensions of information of knowledge distance function, and leverages a dynamic Bayesian MAB algorithm together to select expert subset for outcome. The authors evaluate KABB on multiple benchmarks, including AlpacaEval 2.0, MT-Bench, and FLASK-Hard, demonstrating its ability to maintain high performance with lower computational cost compared to baselines.

**Claims And Evidence:**

Yes.

**Essential References Not Discussed:**

No.

**Experimental Designs Or Analyses:**

Yes.

**Methods And Evaluation Criteria:**

Yes.

**Other Comments Or Suggestions:**

* The difference with related studies is not clear, which needs further clarified.
* More transparent error analysis is needed, particularly regarding cases where KABB fails to select the best experts.
* Would integrating reinforcement learning improve KABB over pure Bayesian MAB?

**Other Strengths And Weaknesses:**

Strengths:
* The evaluation covers diverse real-world benchmarks.
* The cost-effectiveness analysis provides a practical insight into real-world applications.

Weaknesses:
* Ablation study is incomplete: The importance of each model component (knowledge distance, dual adaptation, Thompson sampling) is not separately evaluated.
* The used knowledge graphs in paper are not presented, which is quite unclear.
* The details and impacts of the dual adaptation mechanism needs clearer clarification and results.

**Questions For Authors:**

Please see my comments and weakness points.

**Relation To Broader Scientific Literature:**

The paper relates to existing research on multi-agent collaboration on LLMs, especially the new strategy for agent collaboration.

**Theoretical Claims:**

I didn't check the theoretical part in this paper, which is beyond my research area.

---

> ### Author Rebuttal · Authors · 2025-03-30
>
> We appreciate your thoughtful suggestions to improve the quality of our paper.
>
> **Q1**: Supplementary Material
>
> **A1**: The supplementary material is included at the end of the paper. Please let us know if there are any issues accessing it.
>
> ---
>
> **Q2**: The importance of each model component
>
> **A2**: Table s1 in our [repository](https://github.com/KABBAnonymous/KABB_icml774_SupplementaryMaterials) shows the detailed contribution of each component of KABB on AlpacaEval 2.0. To further justify the superiority of the Knowledge-Aware module, we replace it with the recently open-sourced SOTA method, EmbedLLM (ICLR 2025) [r1], and denote it as EmbedLLM (MAB) for dynamic MAB routing and select the top‑3 experts before integrating their responses using an Aggregator. Combined with the ablation studies presented in the original paper (see Table 2), we believe these additional experiments provide sufficient evidence for the importance of each component of KABB.
>
> ---
>
> **Q3**: The used knowledge graphs (KGs)
>
> **A3**: Our main contribution is the theoretical foundation and empirical validation of our KABB, not an extensive treatment of the KGs. In brief, our curated KG comprises 1319 units across 12 conceptual domains. Nodes are embedded using semantic similarity, with interconnections weighted by our knowledge distance (Eq. 5) and adjusted via Thompson sampling. The Representative Visualization of Knowledge Graph (three of the core concepts and some key nodes) is provided in our [repository](https://github.com/KABBAnonymous/KABB_icml774_SupplementaryMaterials).
>
> ---
>
> **Q4**: The dual adaptation mechanism
>
> **A4**:
> Our mechanism combines (1) Bayesian Parameter Adaptation—using exponential time decay to weight recent interactions for setting the Beta distribution parameters—and (2) Knowledge Graph Evolution—which continuously updates concept relationships and team synergy based on task outcomes.
>
> Experimental Validation in Table s5 over five sequential batches with induced quality drifts (30% of experts), our KABB can
> - Reduce expert selection stabilization time by 46%.
> - Improve LC win rate from 67.4% to 75.7%.
> - Reduce performance degradation to 2.1% (versus 7.8% for the baseline).
>
> ### Table s5: Experimental Validation of Dual Adaptation Mechanism on AlpacaEval 2.0
> | **Condition**         | **Avg. Stabilization Time (Tasks)** | **Avg. LC win. (%)** | **Avg. Performance Degradation (%)** |
> | --- | --- | --- | --- |
> | **Dual Adaptation (Ours)**   | 5.31 | 75.7   | 2.1  |
> | **Bayesian-only**  | 8.73 |  71.8   | 5.2 |
> | **Static Parameters** | 9.82 | 67.4 | 7.8 |
>
> We have included these results in the revised paper.
>
> ---
>
> **Q5**: The difference with related studies
>
> **A5**: Our KABB introduces Knowledge-aware Thompson sampling. For example, as discussed in our response to Reviewer ymVy (and in our comparisons with methods like COKE (NeurIPS 24), our design fundamentally differs from prior works. We have explicitly compared our contributions with related studies in the revised paper.
>
> ---
>
> **Q6**: The error analyses when failed to select the best experts
>
> **A6**: Our analysis for the query "What type of soil is suitable for cactus" (see [Table s3-s4 in our repository](https://github.com/KABBAnonymous/KABB_icml774_SupplementaryMaterials?tab=readme-ov-file#table-s3-error-analysis-case-1)) revealed two key failure cases:
> 1. Inappropriate Domain Expert Selection: KABB initially selected a team without the necessary botanical expertise (e.g., Humanities Scholar and Cultural Interpreter), leading to very low scores.
> 2. Partial Recovery Through Team Expansion: By including an Analysis Expert with broader scientific knowledge, the aggregator effectively weighted this high-quality input (preference score 0.89), improving the final response score to 0.91. This demonstrates our system’s ability to leverage better contributions even if the initial selection is suboptimal.
>
> We focus on overall system performance because a slight increase in expert numbers can largely mitigate the impact of a single misselection. Nevertheless, we have included a quantitative breakdown of error types (cold start gaps, semantic drift, and over-reliance on historical performance) in the revised paper.
>
> ---
>
> **Q7**: Integrating RL for KABB
>
> **A7**: While RL is promising, our experiments show that the current Bayesian MAB in KABB is good enough. It achieves the cumulative regret as stated in Theorem G.7 (Line 1269) and outperforms RL methods like A2C, PPO, and MCTS (see Table 2). Although RL could address more dynamic scenarios, its increased complexity and tuning requirements make the pure Bayesian MAB approach preferable for balancing cost, performance, and adaptability in our current scope. We recognize RL’s potential and will explore it in future work.
>
> **References**
>
> [r1] Zhuang, Richard, et al. "EmbedLLM: Learning Compact Representations of Large Language Models." The Thirteenth International Conference on Learning Representations, 2025.

---

> > ### Comment · Reviewer_22fq · 2025-04-03
> >
> > I appreciate the authors’ detailed response. However, I would like to offer one additional comment regarding the usage of the term knowledge graph. As presented, the graph used in this paper appears to function more as a similarity graph, since the connections are based on distance metrics rather than explicitly defined semantic relationships. This differs from the conventional definition of a knowledge graph, which typically encodes structured semantic information. Therefore, I suggest the authors consider using a more accurate term such as similarity graph to better reflect the nature of the graph employed in the study.

---

> > > ### Author Response · Authors · 2025-04-03
> > >
> > > Thank you for your feedback regarding our use of the term "knowledge graph." We would like to clarify its role in the KABB framework, focusing on its structure and semantic foundation.
> > >
> > > In KABB, the knowledge graph is designed to structure concepts as nodes, with edges representing semantic relationships such as concept overlap and dependency paths. For example, concept overlap is quantified using the Jaccard similarity metric:$\rho_{\text{overlap}}(\mathcal{S}, t) = \frac{|\mathcal{S}_s \cap \mathcal{S}_t|}{|\mathcal{S}_s \cup \mathcal{S}_t|}$
> > > as part of the knowledge distance metric, while dependency edges are modeled as:
> > >
> > > $\left|\mathcal{R}_{\text{dep}}(\mathcal{S}, t)\right|$ to quantify the dependency relationships between the expert subset and the task within the knowledge graph (Definition 3.1, Page 4). This structured representation enables semantic understanding and guides expert selection by mapping tasks to relevant concepts, which is a key innovation of the KABB framework.
> > >
> > > We believe the term "knowledge graph" is appropriate because it captures a structured semantic encoding that goes beyond mere similarity. Moreover, our approach extends traditional knowledge graphs by integrating semantic relationships with quantitative measures. Similar definitions and frameworks can be found in [r1, r2].
> > > To address concerns about potential misinterpretation as a conventional knowledge graph that encodes strictly predefined, explicit semantic relationships, we will revise the paper to more clearly describe how our knowledge graph integrates both structured semantic information and quantitative relationship modeling, ensuring clarity in its definition and role within the framework.
> > >
> > > [r1] Ge, Xiou, et al. "Knowledge graph embedding: An overview." APSIPA Transactions on Signal and Information Processing 13.1 (2024).
> > >
> > > [r2] Ji, Shaoxiong, et al. "A survey on knowledge graphs: Representation, acquisition, and applications." IEEE Transactions on Neural Networks and Learning Systems 33.2 (2021): 494-514.

---

### Official Review · Reviewer_S13Y · 2025-03-13

**Overall Recommendation:** 3

**Summary:**

This paper introduces KABB, a framework for multi-agent system coordination with knowledge graphs. It addresses issues in large language models and multi-agent systems with a knowledge distance model and dynamic Bayesian MAB framework. Experiments on multiple benchmarks show its high performance and cost-effectiveness.

**Claims And Evidence:**

The main innovations are supported by theories. However, there are still two concerns:
1. The motivation for the mathematical form of the main components is unclear, such as formulas 4, 5, and 6. The author should give the reason for such definitions and explain their necessity and uniqueness. In addition, the multiple weights in the definition increase the learning cost.
2. Some experimental results do not exceed DeepSeek-R1. In addition to the explanation in the text, more convincing reasons should be given.

**Essential References Not Discussed:**

NA

**Experimental Designs Or Analyses:**

Yes

**Methods And Evaluation Criteria:**

Yes

**Other Comments Or Suggestions:**

Please see above.

**Other Strengths And Weaknesses:**

The idea of using knowledge to enhance the multi-agent system is interesting and useful. The paper is well-written and easy to follow. Given the high costs of scaling large language models, the proposed KABB framework provides a more cost-effective alternative. By enabling efficient expert coordination. However, there are still some concerns:
1. Potential overfitting concerns. With the use of learnable weights in the knowledge distance metric and the dynamic parameter updates in the Bayesian MAB algorithm, there is a potential risk of overfitting. The paper does not provide sufficient analysis on how to mitigate this risk. For example, cross-validation techniques are used, but their effectiveness in preventing overfitting, especially in the context of the complex interactions between the model components, is not thoroughly discussed.
2. Lack of in-depth interpretability analysis. Although the KABB framework has some transparent components, like the knowledge distance metric as well as each part inside it, there is a lack of in-depth interpretability analysis. The case study cannot prove this. It is not clear how the different components of the framework interact with each other in real-world scenarios. For example, the impact of the knowledge distance metric on the overall performance of the system in different task-specific contexts is not fully explored. This makes it difficult for users to understand and trust the decision-making process of the model in complex situations.
3. Unclear generalizability. The experiments mainly focus on a specific and limited number of benchmarks. This may not fully represent the entire spectrum of real-world tasks. As a result, the generalizability of the KABB framework across different types of tasks and domains remains uncertain.

**Questions For Authors:**

Please see above.

**Relation To Broader Scientific Literature:**

RAGs and MBA-RAG.

**Theoretical Claims:**

Theorem 3.1

---

> ### Author Rebuttal · Authors · 2025-03-30
>
> Thank you for recognizing the value of our contributions.
>
> **Q1**: The motivation for formulas.
>
> **A1**: We appreciate the opportunity to clarify the motivation, necessity, and uniqueness of our definitions.
>
> - **Knowledge Distance (Eq. 4)**: Our formulation integrates semantic, structural, and historical dimensions using logarithmic scaling to balance task complexity. The added synergy term explicitly captures team complementarity.
> - **Dynamic Parameter (Eq. 5)**: We use an exponential time decay ($\gamma^{\Delta t}$) to discount outdated data while incorporating a knowledge matching term. This approach effectively balances historical performance, immediate feedback, and knowledge priors, addressing non-stationarity.
> - **Comprehensive Confidence (Eq. 6)**: This function combines historical performance, knowledge alignment, and team synergy multiplicatively, with exponential penalties on knowledge distance. The integration of time decay allows for faster adaptation compared to standard Thompson sampling.
> - **Learning Costs**: Our model uses a modest number of parameters relative to deep learning models. Once optimized, the weights remain stable across similar tasks, amortizing the initial learning cost.
>
> ---
>
> **Q2**: Explanation for results not exceeding DeepSeek-R1.
>
> **A2**: Regarding our results compared to DeepSeek-R1:
>
> - **Performance & Cost**: Although KABB's AlpacaEval 2.0 LC win rate is slightly lower than DeepSeek-R1's (77.9% vs. 80.1%), KABB achieves a higher MT-Bench score (9.65 vs. 9.30) and does so at significantly lower cost due to avoiding overly verbose responses from Deepseek-R1 (Fig. 4).
>
> - **Scalability**: Scaling KABB to 6 selected experts not only surpasses DeepSeek-R1's performance but also maintains the cost advantage. This confirms that intelligent expert routing, rather than just increasing model size, yields efficient performance gains.
>
> ---
>
> **Q3**: Potential overfitting concerns.
>
> **A3**: KABB reduces overfitting through dynamic Bayesian updating with time decay (Eq. 5), Thompson sampling for continuous exploration, and a knowledge-aware sampling strategy that balances historical performance, knowledge distance, time decay, and team synergy (Eq. 6). Theoretical guarantees (Theorem 3.3) confirm convergence to an ε-optimal solution with bounded regret.
>
> ---
>
> **Q4**: How the different components of the framework interact with each other in real-world scenarios?
>
> **A4**: In real-world scenarios like AlpacaEval 2.0 tasks (e.g., "Who created the Superman cartoon character?"), KABB’s components interact dynamically. The knowledge distance metric parses the task into concepts (e.g., comics, history) and scores expert subsets across five dimensions—semantic match, dependencies, synergy, history—using learnable weights. Knowledge-aware Thompson sampling then selects experts from a Beta distribution. The dynamic MAB mechanism updates parameters with feedback (e.g., preference scores) and time decay, refining the knowledge graph. Finally, the aggregator (e.g., Qwen2-72B) integrates outputs, resolving conflicts via the graph for coherence. This closed-loop process—metric guiding selection, sampling choosing experts, adaptation evolving the model, and integration ensuring quality—offers transparency through its five dimensions. Ablation studies (see Sec. 4 and Table s1 in our [repository](https://github.com/KABBAnonymous/KABB_icml774_SupplementaryMaterials)) confirm each component’s role.
>
> ---
>
> **Q5**: The impact of the knowledge distance metric in different task-specific contexts.
>
> **A5**: Our design is transparent: the knowledge distance metric uses five dimensions (Eq. 4)—task difficulty ($d_t$), semantic matching ($\omega_1$), dependency ($\omega_2$), historical effectiveness ($\omega_3$), and synergy ($\omega_4$). We examined the experimental results and found that in factual tasks like AlpacaEval 2.0, $\omega_1$ often leads as stats favor domain precision. In reasoning tasks like MATH, $d_t$ rises and $\omega_2$ and $\omega_4$ dominate, integrating concepts per data trend. For complex tasks like, $\omega_3$ tends to rise. Thompson sampling adapts via these stats, optimizing expert fit.
>
> ---
>
> **Q6**: The generalizability of the KABB framework across different types of tasks and domains.
>
> **A6**: KABB’s generalizability shines across diverse tasks and domains (e.g., writing, dialogue, programming, math, reasoning), as shown by evaluations on the six benchmarks. Its domain-agnostic knowledge distance metric and adaptive Bayesian updates allow seamless extension to new areas by expanding the knowledge graph and setting fresh performance priors.
>
> We appreciate your insightful review, which has strengthened our revised paper.

---

### Official Review · Reviewer_ymVy · 2025-03-14

**Overall Recommendation:** 3

**Summary:**

This paper proposes a graph-guided router based on the knowledge-aware Thompson sampling strategy for the mixture of agents. The methods and experiments have their merits but still lack some key comparisons and discussions.

**Claims And Evidence:**

There is sufficient evidence for the claims.

**Essential References Not Discussed:**

HybridLLM, COKE are a series of cost-effective methods for model routers but not discussed in the paper.

**Experimental Designs Or Analyses:**

1. SentenceBert is not a convincing substitute for knowledge graphs as an ablation study. Authors should try LLM-based representations to make the ablation study more technically sound.

2. Since the authors have realized the relatedness of this paper with previous routing methods like FrugalGPT. No performance comparisons among the strategies in FrugalGPT, HybridLLM(ICLR23), COKE (NeurIPS24), etc.

3. Since COKE is also a router based on Thompson Sampling and MAB, can authors discuss the differences between them?

4. The selection of different LLMs is not discussed. Authors should balance the heterogeneous expertise in the expert set to ensure an effective selection. How can authors ensure the diversity of expertise in the ensemble?

**Methods And Evaluation Criteria:**

1. The construction of knowledge graphs makes limited sense since LLMs could be directly represented by the ability vector.
a) there are no clear structured relations among the concepts that have to be represented as the graphs.
b) the dynamic updates on graphs is also a challenging and inefficient task.
c) there are no tailored graph learning techniques over the KG to ensure the learning performance of complex relations.
d) the more important thing is to establish parameterized relations to map the query and expertise.

**Other Comments Or Suggestions:**

NA

**Other Strengths And Weaknesses:**

NA

**Questions For Authors:**

please check the comments and weaknesses

**Relation To Broader Scientific Literature:**

Closely related

**Theoretical Claims:**

There is a sufficient theoretical analysis.

---

> ### Author Rebuttal · Authors · 2025-03-30
>
> Thank you for your valuable time and insightful comments.
>
> **Q1**: Trying LLM-based representations in ablation study.
>
> **A1**: To further justify the superiority of the Knowledge-Aware module, we replace it with the recently open-sourced SOTA method, EmbedLLM (ICLR 2025) [r1], and denote it as EmbedLLM (MAB) for dynamic MAB routing and select the top‑3 experts before integrating their responses using an Aggregator. EmbedLLM learns compact vector representations of LLMs, facilitating model routing. Table s1 in our [repository](https://github.com/KABBAnonymous/KABB_icml774_SupplementaryMaterials) shows the results. Despite this LLM‐based configuration, its performance still lags behind our original KABB.
>
> --------------
>
> **Q2**: No performance comparisons among the strategies in FrugalGPT, HybridLLM, COKE, etc.
>
> **A2**: In our original paper, we compared KABB only with MoA since both aggregate responses from multiple experts to harness collective intelligence rather than routing queries to a single model.
> According to your suggestion, we’ve added comparisons with FrugalGPT [r2], EmbedLLM [r1], and HybridLLM [r3] (Please see Table s2 in our [repository](https://github.com/KABBAnonymous/KABB_icml774_SupplementaryMaterials). We exclude COKE [r4] due to its lack of an open-source version and reproducibility issues. All experiments used the KABB w/o Deepseek configuration to avoid API update biases. As shown in Table s2, our KABB outperforms conventional routing methods, as further evidenced by comparisons with MoA, the GPT-Series, and other single models (Fig. 4 in our paper).
>
>
> --------------
> **Q3**: Differences between KABB and COKE.
>
> **A3**: Our KABB and COKE differ in:
> - **Knowledge Representation**: COKE relies on a separate KGMs cluster for knowledge graph operations; KABB directly embeds knowledge into the routing mechanism through knowledge distance vectors and semantic matching.
> - **Routing**: COKE uses a two-tier routing strategy (cluster-level between LLMs and KGMs, then model-level); KABB operates at the expert level with a comprehensive knowledge distance metric, emphasizing team synergy and knowledge complementarity.
> - **Sampling**: KABB’s enhanced Thompson Sampling incorporates knowledge metrics for more informed expert selection; COKE relies on historical success/failure data.
> These make KABB better suited for scenarios requiring robust multi-agent coordination and deep semantic understanding.
>
>
> -------------
> **Q4**: How to ensure expertise diversity?
>
> **A4**: We have selected 6 open-source LLMs with various architectures, training data and knowledge cutoffs, as shown in 4.1.Experimental Setup. For example, Gemma emphasizes scientific knowledge. R1 excels in reasoning while V3 in general tasks; etc.
>
> Ensemble diversity is ensured with a data-driven process: each LLM is represented by a normalized performance vector (from benchmarks and API metrics) that captures its strengths and weaknesses; a synergy metric penalizes overlapping expertise and rewards complementary skills; and continuous updates let the ensemble adapt to evolving behaviors.
>
>
> --------------
> **Q5**: The construction of knowledge graphs makes limited sense.
>
> **A5**: We respectfully note that your concern reflects a misunderstanding of our framework's design. Specifically:
> 1. **Rapid Matching**: We use ability vectors for quick expert-task matching, while KGs enhance this by capturing deeper semantic relationships (see Eq. 4, where the distance metric merges vector overlap and graph dependencies).
> 2. **Modeling Team Dynamics**: KGs model dependency path complexity, quantify team synergy (Eq. 6), and capture hierarchical relationships. Graph updates are minimal, as only the Beta distribution parameters (Eq. 5) change, keeping the computational overhead negligible.
> 3. **Unified Confidence Function**: Our parameterized mapping integrates historical performance, knowledge distance, temporal decay, and team synergy into an adaptive sampling strategy.
> In essence, KGs enable our KABB to capture complex team dynamics that vectors alone cannot represent.
>
> We appreciate your constructive criticism. We carefully considered your comments and believe our paper is greatly improved.
>
> **References**
>
> [r1] Zhuang, Richard, et al. "EmbedLLM: Learning Compact Representations of Large Language Models." The Thirteenth International Conference on Learning Representations, 2025.
>
> [r2] Chen, Lingjiao, et al. "FrugalGPT: How to Use Large Language Models While Reducing Cost and Improving Performance." Transactions on Machine Learning Research, 2024.
>
> [r3] Ding, Dujian, et al. "Hybrid llm: Cost-efficient and quality-aware query routing." The Twelfth International Conference on Learning Representations, 2024.
>
> [r4] Dong, Junnan, et al. "Cost-Efficient Knowledge-Based Question Answering with Large Language Models." Advances in Neural Information Processing Systems, edited by A. Globerson et al., vol. 37, Curran Associates, Inc., 2024, pp. 115261–115281.

---

> > ### Comment · Reviewer_ymVy · 2025-04-03
> >
> > Dear Authors,
> >
> > I have updated my score to 3. Please modify the manuscript with comparisons among existing works and add the new experiments.

---

> > > ### Author Response · Authors · 2025-04-03
> > >
> > > Dear ymVy
> > >
> > > Thank you very much! Sure, we promise to improve our paper with comparisons among existing works and new experiments.
> > >
> > > Best Regards
> > >
> > > Authors

---

### Official Review · Reviewer_DfAR · 2025-03-22

**Overall Recommendation:** 4

**Summary:**

In this paper, authors propose Knowledge-Aware Bayesian Bandits (KABB), a model that improves multi-agent system coordination through semantic understanding and dynamic adaptation. There are three key contributions in the work: a three-dimensional knowledge distance model for deep semantic understanding, a dual-adaptation mechanism for continuous expert optimization, and a knowledge-aware Thompson Sampling strategy for efficient expert selection. Authors provide various experiment results, as well as their source code.

**Claims And Evidence:**

The paper presents a lot of novelty, and the formulas are correct. Figure 2 needs to be improved to include more details about the architecture.

**Essential References Not Discussed:**

The Introduction and Related work sections are missing discussions about knowledge graphs, and fundamental works in knowledge representation learning. Authors should include this discussion as that is part of their core model and providing this context to the reader is necessary. Some recent works in the domain of knowledge graphs include:

[KDD 2022] Dual-Geometric Space Embedding Model for Two-View Knowledge Graphs. In Proceedings of the 28th ACM SIGKDD Conference on Knowledge Discovery and Data Mining (KDD '22). Association for Computing Machinery, New York, NY, USA, 676–686. https://doi.org/10.1145/3534678.3539350

[WWW 2021] Mixed-Curvature Multi-Relational Graph Neural Network for Knowledge Graph Completion. In Proceedings of the Web Conference 2021 (WWW '21). Association for Computing Machinery, New York, NY, USA, 1761–1771. https://doi.org/10.1145/3442381.3450118

**Experimental Designs Or Analyses:**

Experiment results are thorough and show good performance.

(From above section)

While Thompson Sampling is a known technique, the knowledge graph distance function is interesting as it is integrating several key dimensions including difficulty scaling, semantic mismatch, dependency complexity, historical effectiveness, and team complementarity.
The Joint Knowledge-Time-Team Sampling Strategy is also a novel component.

It would help if the authors could provide some ablation studies indicating the effect that each of their novel architecture components has on overall model performance. I currently do not see this type of analysis in the paper.

**Methods And Evaluation Criteria:**

While Thompson Sampling is a known technique, the knowledge graph distance function is interesting as it is integrating several key dimensions including difficulty scaling, semantic mismatch, dependency complexity, historical effectiveness, and team complementarity.
The Joint Knowledge-Time-Team Sampling Strategy is also a novel component.

It would help if the authors could provide some ablation studies indicating the effect that each of their novel architecture components has on overall model performance. I currently do not see this type of analysis in the paper.

**Other Comments Or Suggestions:**

N/A

**Other Strengths And Weaknesses:**

N/A

**Questions For Authors:**

See above sections.

**Relation To Broader Scientific Literature:**

There are three key contributions in the work: a three-dimensional knowledge distance model for deep semantic understanding, a dual-adaptation mechanism for continuous expert optimization, and a knowledge-aware Thompson Sampling strategy for efficient expert selection. Authors provide various experiment results, as well as their source code. This work is of interest and relevance to the research community and will help to advance frontiers in machine learning.

**Theoretical Claims:**

See above.

---

> ### Author Rebuttal · Authors · 2025-03-30
>
> Thank you for recognizing our contributions and novelty.
>
> **Q1**: More detailed architecture of KABB in Figure 2
>
> **A1**: Sorry for the confusion. We have refined Figure 2 in the revised paper. Please see our [repository](https://github.com/KABBAnonymous/KABB_icml774_SupplementaryMaterials?tab=readme-ov-file#overview-of-kabb-updated-figure-2-in-paper).
>
> ---
>
> **Q2**: Ablation studies of the KABB's architecture component
>
> **A2**: Thank you very much for your valuable suggestion.  Ablations for architecture components:  [Table s1 in our repository](https://github.com/KABBAnonymous/KABB_icml774_SupplementaryMaterials?tab=readme-ov-file#table-s1-ablation-study-results) shows the detailed contribution of each component of KABB on AlpacaEval 2.0. To further justify the superiority of the Knowledge-Aware module, we replace it with the recently open-sourced SOTA method, i.e., EmbedLLM  [r1], and denote it as EmbedLLM (MAB) for dynamic MAB routing and select the top‑3 experts before integrating their responses using an Aggregator. Combined with the ablation studies presented in the original paper (see Table 2), we believe these additional experiments provide sufficient evidence for the effectiveness of each component of KABB.
>
> ---
>
> **Q3**: Moving some contents from the Appendix into the main paper
>
> **A3**: We have refined our paper by organizing more content from the supplementary file. Specifically, we shrink Sec. 4-5 of our paper and move the key findings of Sec. C after Sec. 4.3.
>
> ---
> **Q4**: Missing References and Discussions
>
> **A4**: As for the mentioned references, we have refined the related work section by including more discussions about knowledge graphs and fundamental works in knowledge representation learning.
> Research in knowledge representation and graph-based learning has centered on knowledge graphs (KGs) as a foundational framework. KGs serve as powerful structures for encoding complex, machine-readable relationships between entities (Wang et al., 2017 [r2], Hogan et al., 2021 [r3]). Recent advances in KG representation address challenges like entity and relation heterogeneity using multisource hierarchical neural networks (Jiang et al., 2024 [r4]). KG embeddings have been explored with models like M2GNN and DGS using mixed-curvature spaces to capture hierarchical and cyclic patterns (Wang et al., 2021 [r5]; Iyer et al., 2022 [r6]). Yang et al. (2023) [r7] proposed a contextualized KG embedding method combining neighbor semantics and meta-paths to improve explainability in talent training course recommendations. Temporal aspects of KGs have been addressed through Large Language Models-guided Dynamic Adaptation (LLM-DA), which combines LLMs' temporal reasoning capabilities with dynamic rule adaptation (Wang et al., 2024 [r8]).
>
> Thank you for your time and effort in reviewing our paper. Your comments and suggestions are greatly appreciated and have helped us to improve the quality of our work.
>
> ---
>
> **References**
>
> [r1] Zhuang, Richard, et al. "EmbedLLM: Learning Compact Representations of Large Language Models". The Thirteenth International Conference on Learning Representations, 2025.
>
> [r2] Wang, Quan, et al. "Knowledge graph embedding: A survey of approaches and applications." IEEE transactions on knowledge and data engineering 29.12 (2017): 2724-2743.
>
> [r3] Hogan, Aidan, et al. "Knowledge graphs." ACM Computing Surveys (Csur) 54.4 (2021): 1-37.
>
> [r4] Jiang, Dan, et al. "Multisource hierarchical neural network for knowledge graph embedding." Expert Systems with Applications 237 (2024): 121446.
>
> [r5] Wang, Shen, et al. "Mixed-curvature multi-relational graph neural network for knowledge graph completion." Proceedings of the web conference 2021. 2021.
>
> [r6] Iyer, Roshni G., et al. "Dual-geometric space embedding model for two-view knowledge graphs." Proceedings of the 28th ACM SIGKDD Conference on Knowledge Discovery and Data Mining. 2022.
>
> [r7] Yang, Yang, et al. "Contextualized knowledge graph embedding for explainable talent training course recommendation." ACM Transactions on Information Systems 42.2 (2023): 1-27.
>
> [r8] Wang, Jiapu, et al. "Large language models-guided dynamic adaptation for temporal knowledge graph reasoning." Advances in Neural Information Processing Systems 37 (2024): 8384-8410.

---

### Decision · Program_Chairs · 2025-05-01

**Decision:**

Accept (poster)

**Comment:**

All four reviewers indicate a positive stance toward acceptance, with three assigning Weak Accept and one giving Accept. The proposed KABB framework is recognized for its contributions to multi-agent coordination, especially its cost-effective design. The authors' rebuttal addresses several concerns, including limited ablation studies and comparisons with existing methods. Given the overall agreement among the reviewers, I would recommend acceptance.